

# Time-dependent Probabilistic Tsunami Hazard Analysis for Western Sumatra, Indonesia, Using Space-Time Earthquake Rupture Modelling and Stochastic Source Scenarios

Ario Muhammad[1,2], Katsuichiro Goda[3], Maximilian J. Werner[2]

[1]Civil Engineering Department, President University, Bekasi, 17530, Indonesia
[2]School of Earth Sciences, University of Bristol, Bristol BS8 1TR, UK
[3]Department of Earth Sciences, Western University, London, N6G 2V4, Canada

*Correspondence to*: Ario Muhammad (ario.muhammad@president.ac.id)

**Abstract.** We develop a novel framework of time-dependent probabilistic tsunami hazard analysis (PTHA) and apply it to
Western Sumatra, Indonesia, where future tsunamigenic events are anticipated in the Mentawai region of the Sunda
subduction zone. An earthquake rupture model taking into account the spatiotemporal interaction of major megathrust
segments is used to simulate future tsunamigenic earthquakes. The earthquake rupture process of the segments is
characterized by a multivariate Bernoulli model with interarrival times following a Brownian passage-time distribution and
the dependency between segments specified by a spatial correlation function. We calibrate this model with historical
ruptures of the Mentawai thrust in the last 450 years. A total of ≥100,000 time-dependent earthquake rupture cases are then
coupled with a stochastic tsunami simulation method to evaluate tsunami hazards. We generate a total of 6,300 stochastic
tsunami source models from six magnitude scenarios between M 7.75 and M 9.0 and obtain time-dependent PTHA results
for seven different periods (1, 5, 10, 20, 30, 50 and 450 years). We further compare the time-dependent PTHA results with a
time-independent PTHA approach to investigate the influence of the spatiotemporal earthquake rupture model. The space-
time interaction model successfully generates annual seismic moment rates consistent with the observations. Moreover, the
model can capture the uncertainty of future time-dependent tsunami hazards. On the other hand, the time-independent
approach produces slightly higher hazard estimates than the time-dependent model for long-term hazard assessments (> 450
years).

## 1. Introduction

Massive tsunamigenic earthquakes have occurred in recent years and caused devastating effects in tsunami-prone areas.
Globally, in the last 50 years, seventeen significant tsunami earthquakes occurred, claiming a total economic loss exceeding
$14 trillion and almost 290,000 deaths (USGS, 2020). Two notable tsunamigenic events contributing to these significant
economic losses and fatalities were the 2004 Aceh-Andaman tsunami in Indonesia (Rossetto et al., 2007) and the 2011
Tohoku-Japan tsunami (Raby et al., 2015). Over the next decades, major tsunamigenic events are anticipated in several



subduction zones. These include the Mexican subduction zone (Mori et al., 2017), the Nankai-Tonankai Trough (Goda et al., 2020), the Makran subduction zone (Momeni et al., 2020), the Hikurangi subduction zone (Wallace et al., 2016), the Philippines subduction zone (Løvholt et al., 2012), the Cascadia subduction zone (Park et al., 2017), and the Mentawai region of the Sunda subduction zone (Muhammad et al., 2017). Therefore, comprehensive tsunami risk mitigation in these tsunami-prone areas is essential.

35         A fundamental requirement to assess tsunami risk and determine effective tsunami risk mitigation strategies is Probabilistic Tsunami Hazard Analysis (PTHA). In general, PTHA can be carried out in four stages: (1) earthquake occurrence modelling and rupture area characterization, (2) tsunami source generation, (3) tsunami propagation and inundation, and (4) tsunami hazard estimation. Regarding earthquake occurrence modelling, the current state-of-the-art approach employs a memory-less Poisson process for long-term hazard assessment and is commonly adopted in previous
PTHA studies (Burroughs and Tebbens, 2005; Tinti et al., 2005; Orfanogiannaki and Papadopoulos, 2007). However, assuming a lack of memory between major earthquake occurrences is often viewed as a first approximation, inconsistent with the physics of elastic rebound (Reid, 1911; Anagnos and Kiremidjian, 1984; Berryman et al., 2012). As a result, many have adopted a renewal model of earthquake occurrence (Matthews et al., 2002; Zhuang et al., 2012; Field et al., 2014) to carry out a time-dependent analysis, specifically for probabilistic seismic hazard analysis (PSHA; Akinci et al., 2005; Field
et al., 2014). Recently, Goda (2019) and Fukutani et al. (2021) applied this approach to PTHA. In addition, Ceferino et al. (2020) developed a new space-time earthquake occurrence model by characterizing spatiotemporal interactions of major tsunamigenic events as a correlated, multivariate Bernoulli process of neighboring segments each obeying marginal Brownian passage-time (BPT) recurrence distributions. Their model includes spatial correlations between ruptures and yet guarantees prescribed marginal distributions on each segment via copulas.

50         Furthermore, recent tsunami hazard assessments have progressed significantly in earthquake source modelling. For instance, stochastic earthquake source modelling considering the uncertainty and dependency of earthquake source parameters has been adopted to estimate the tsunami hazard in tsunami-prone regions (e.g. Mori et al., 2017; Park et al., 2017; Momeni et al., 2020). Recent work has also used high-resolution spatial grids to simulate tsunamis to produce more accurate tsunami hazard results (e.g. <90 m; Griffin et al., 2016; Goda et al., 2020). These recent advances in stochastic
source modelling should be integrated into PTHA.

In this study, we develop a novel time-dependent PTHA framework that combines a space-time earthquake interaction model with stochastic source modelling and high-resolution tsunami simulations for the Mentawai region of the Sunda subduction zone in Indonesia. We implement the space-time earthquake occurrence model developed by Ceferino et al. (2020) to characterize the occurrence of future earthquake ruptures. This approach models ruptures on individual
segments with temporal renewal processes and prescribes spatial correlation between ruptures in neighboring segments to enable multi-segment ruptures. To characterize realistic earthquake rupture patterns for assessing tsunami hazard, we incorporate stochastic earthquake source scenarios, including uncertainties in the size of the rupture plane and spatial heterogeneity of earthquake slip (Muhammad et al., 2016). In other words, the proposed time-dependent earthquake rupture





modelling method combines the spatiotemporal interaction earthquake occurrence model by Ceferino et al. (2020) and the
stochastic-source PTHA method by Muhammad et al. (2016) to enable time-dependent PTHA based on segmented
earthquake rupture sources. The proposed framework is then implemented to carry out PTHA due to future tsunamigenic
events generated in the Mentawai region. Since the time-dependent hazard estimation leads to more realistic short-term
results (e.g. < 50 years) compared to long-term assessments with a Poisson process, adopting the new time-dependent PTHA
approach is more appropriate for the Mentawai-Sunda zone, where major tsunamigenic events are impending over the next
few decades based on paleogeodetic and paleotsunami studies (Philibosian et al., 2014, 2017).

In a national context, PTHA for Indonesia was developed by Horspool et al. (2014). However, limitations of the
current tsunami hazard assessment in Indonesia include: (1) a time-independent Poisson process of earthquake occurrence
was adopted, which is inconsistent with time-dependent hazard estimation over a short period, (2) a uniform slip of
earthquake rupture was used, which could underestimate tsunami hazard (e.g. Muhammad et al., 2018), and (3) coarse grids,
i.e. ~1,800 m in the coastal region and ~5,400 m in deep ocean areas (below 1,000 m), were implemented for tsunami
simulation, without performing detailed run-up simulations. By adopting a finer tsunami simulation grid (50 m) together
with more realistic heterogeneous earthquake slip models, we improve the prior work by Horspool et al. (2014) by
accounting for the variability and uncertainty of the earthquake source models as well as the time-dependent earthquake
occurrence.

To carry out the analysis, we first collect supercyclic earthquake data since the 16[th] century (a 450-year period) in
the Mentawai region from paleogeodetic measurements (Natawidjaja et al., 2006; Sieh et al., 2008; Philibosian et al., 2014,
2017). The past fault rupture areas can be used to construct the potential fault rupture areas of the earthquake occurrence
model. The term supercyclic refers to a long-term cluster of differently sized megathrust earthquakes leading to the final
complete failure across multiple segments of a subduction zone (Herrendörfer et al., 2015). Second, based on the fault
rupture areas and occurrence times of the past earthquakes, we build a space-time interaction model of the large earthquake
ruptures (Ceferino et al., 2020). Third, the defined fault rupture area is further divided into several segments representing six
magnitude ranges, namely M 7.75, 8.0, 8.25, 8.5, 8.75, and 9.0, which we use to define the rupture areas for the stochastic
earthquake source generation (Muhammad et al., 2016). Fourth, we simulate stochastic tsunamis with a fine grid of 50 m by
solving nonlinear shallow-water wave equations (Goto et al., 1997). We target the coastal areas of three key cities, namely
Pariaman, Padang, and Painan. Fifth, we model earthquake rupture with the space-time interaction approach over seven
periods, namely 1, 5, 10, 20, 30, 50 and 450 years. In this study, we mostly focus on the short period (<50 years), whereas
the 450 years is considered as a representation of a longer period and corresponds to the period of the past earthquake data
used in the study area (i.e. from 1597 to 2010). Finally, we derive site-specific tsunami hazard curves and tsunami hazard
maps (i.e. maximum tsunami wave heights).

The paper is organized as follows. Section 2 presents the new methodology for time-dependent PTHA, including
the time-independent and time-dependent models (section 2.1), earthquake rupture modelling with space-time interaction
(section 2.2), and the stochastic earthquake source generation and tsunami simulation (section 2.3). Section 3 presents results



and discussion of the Bayesian parameter estimation of the occurrence model (section 3.1), earthquake rupture modelling (section 3.2), and time-dependent PTHA (section 3.3). This section also compares the hazard curves and maps of the time-dependent PTHA with the time-independent approach.

## 2. Methodology

### 2.1. PTHA framework

#### 2.1.1. Time-independent formulation

Time-independent PTHA assumes that tsunamigenic earthquake occurrence follows a Poisson process. The probability of observing the occurrence of a maximum tsunami height $H$ equal to or greater than a specific value $h$ in $t$ years and at a specific location is:

$$P(H \geq h|t) = 1 - \exp\left[-\lambda(H \geq h) \cdot t\right] \tag{1}$$

where $\lambda(H \geq h)$ is the mean annual rate of $H \geq h$. The term tsunami height used in this paper is the height of water flow above the mean sea level. Moreover, the tsunami depth refers to the height of water flow above ground level. Subtracting the ground elevation from the tsunami wave height at the place of interest yields the depth.

The mean annual rate $\lambda(H \geq h)$ can be expressed as a filtered Poisson process (Parsons and Geist, 2009):

$$\lambda(H \geq h) = \sum_{i=1}^{N_{sources}} \lambda_i(M \geq M_{min}) \cdot \int P_i(H \geq h|\theta) \cdot S(\theta|M) \cdot f_i(M) \cdot dM \tag{2}$$

where $N_{sources}$ is the number of tsunami sources and $\lambda_i(M \geq M_{min})$ is the mean annual rate of earthquakes larger than $M_{min}$ of the $i$th source. The term $P_i(H \geq h|\theta)$ represents the probability of a specific hazard measure $H$ exceeding a pre-defined value $h$ at a location for given tsunami source parameters ($\theta$). $f_i(M)$ is the magnitude-frequency distribution characterizing the $i$th source, whilst $S(\theta|M)$ is the distribution of the uncertain source parameters conditioned on the earthquake magnitude. The frequency-magnitude distribution can be defined by a truncated Gutenberg-Richter (GR) relationship:

$$F(M) = \frac{1 - 10^{-b \cdot (M - M_{min})}}{1 - 10^{-b \cdot (M_{max} - M_{min})}}, \quad M_{min} \leq M \leq M_{max} \tag{3}$$

where the $b$ value in Equation (3) is estimated from the seismicity data of the Advanced National Seismic System (ANSS) catalog (https://earthquake.usgs.gov/data/comcat/; see Figure 1A). Figure 1B shows the $b$ value estimated from seismicity data between 1970 and 2015 in the Mentawai region of the Sunda subduction zone using the maximum likelihood estimation method (Aki, 1965).

To apply stochastic earthquake sources to PTHA, De Risi and Goda (2017) approximated the continuous $f(M)$ with a discrete magnitude function $P(M_j)$ ($j = 1, \dots, n$) of $n$ bins with bin width $\Delta M$ (Figure 1D). The discrete probability can be calculated from

$$P(M_i) = F(M_i + 0.5 \cdot \Delta M) - F(M_i - 0.5 \cdot \Delta M). \tag{4}$$

Hence, we can replace Equation (2) integral with a summation over discrete magnitude bins. Moreover, the mean annual rate $\lambda_i(M \geq M_{min})$ of seismic events greater than $M_{min}$ can be calculated from the fitted GR curve shown in Figure 1C. In this





study, we consider an $M_{min}$ of 7.625 and a $\Delta M$ of 0.25 with six central magnitude values are M 7.75, M 8.0, M 8.25, M 8.5,
M 8.75, and M 9.0. This minimum value is suitable for PTHA because the tsunami hazards from earthquakes smaller than M
7.625 is rarely significant (Muhammad et al., 2016).

### 2.1.2. Time-dependent algorithm

The PTHA framework can be modified to include a renewal occurrence model for time-dependent PTHA. The recurrence of
tsunamigenic events is assumed to be quasi-periodic and depends on the time since the last earthquake. The lognormal and
the BPT distribution have been widely used to describe seismic recurrences for PSHA (Convertito et al., 2012; Field et al.,
2014; Ceferino et al., 2020). We thus adopt the BPT distribution (Matthews et al., 2002) to estimate the (marginal)
probability of tsunamigenic inter-arrival times. Moreover, we include space-time interaction in the time-dependent PTHA
via a spatially correlated rupture model.

Figure 2 shows our approach to conduct time-dependent PTHA. This section presents a brief overview, while
subsequent sections describe individual phases in detail. First, the earthquake source zone is defined and discretized into
segments, each representing the minimum possible ruptured area corresponding to an M7.625 event (see red line arrows in
Figure 2A). The source zone and segments are developed from historical and instrumental data of large earthquakes in the
Mentawai-Sunda zone in the last 450 years (see Figure 2A). Second, using this source geometry model, we generate
earthquake ruptures (see Figure 2B) and stochastic source scenarios (see Figure 2C), considering the uncertainty and
dependency of earthquake source parameters. Crucially, ruptures are time-dependent and spatially correlated. Third, we
simulate tsunami propagation and inundation based on the stochastic earthquake source models to obtain the tsunami hazard
level in the target region (see Figure 2C). Fourth, we determine the probability $P(H \geq h|t)$ of a hazard metric $H$ (i.e.
tsunami height) exceeding a specific $h$ value at a given location over a given period $t$ (see Figure 2D).

The hazard curves $P(H \geq h|t)$ are calculated by combining the rupture modelling and the tsunami simulation
results. As described in more detail below (see section 2.2.1), the occurrence model produces earthquake catalogs with 21
distinct geometrical rupture scenarios spanning six discrete magnitude levels, namely M 7.75, M 8.0, M 8.25, M 8.5, M 8.75,
and M 9.0. Each event may rupture one or multiple segments depending on the magnitude. For example, M 7.75 and M 9.0
events rupture one and five segments, respectively. Furthermore, for each of those 21 rupture scenarios, we generate 300
stochastic source models, which set initial conditions for the corresponding tsunami inundation simulations (see section 2.2.1
and section 2.3) to obtain a total of 6,300 tsunami simulations at 50 m grid resolution in coastal areas.

Figure 3 shows the procedure to develop the final hazard curve for the time-dependent model. First, we select a total
number of simulated earthquake catalogs $N_{sim}$. Second, three results from each earthquake rupture catalog and from the
stochastic tsunami simulations are obtained, namely (1) the simulated earthquake catalogs, (2) the number of occurred
scenarios ($N_{sce}$), and (3) the stochastic simulated heights corresponding to the occurred scenarios. Note that $N_{sce}$ counts the
number of events (i.e. if the same scenario occurs twice in a $t$-year catalog, then they are counted as two events). Based on





the occurred scenarios, the stochastically simulated heights are obtained for each occurred scenario, randomly chosen from a total of 300 simulated heights (i.e. the number of stochastic simulations for each scenario is 300). This height is then stored for each occurred scenario. Those stored heights are further used to define the maximum height for each catalog (i.e. one height for one simulation catalog). The above procedure is then repeated for all rupture catalogs and hence, a total of $N_{sim}$

maximum heights are stored in each $t$-year simulation. These height data are finally used to generate the mean hazard curve using the Complementary Cumulative Distribution Function (CCDF). Thus, only one final hazard curve for each $t$ period is generated.

To determine a suitable value of $N_{sim}$, we first examine the stability of the simulated catalogs by calculating the relative frequency of each scenario as a function of increasing $N_{sim}$. The relative frequency is the ratio of the occurrences of

a scenario to the total number of occurred events in a target period of simulation (e.g. 1 year). A total of 100,000 earthquake rupture simulations are carried out for all seven periods i.e. 1, 5, 10, 20, 30, 50, and 450 years, with a starting simulation time of 2021 (i.e. $year\ 0 = 2021$). The results confirm that $N_{sim} = 100,000$ catalogs are sufficient to produce a stable result and hence, we use $N_{sim} \geq 100,000$, varied for different periods. From an earthquake-tsunami mitigation perspective, the return period of several thousand (e.g. 2,500 years) is generally relevant for tsunami hazard mapping purposes. Moreover, the

tsunami hazard at a lower probability level, i.e. $4 \times 10^{-4}$ and $2 \times 10^{-3}$, may be used to evaluate the tsunami hazard for 1 year and 5 years, respectively (i.e. the return period at these two lower probability levels is 2,500 years). Therefore, we adopt 10,000,000 catalogs for 1-year earthquake rupture simulations to obtain sufficient results at the probability level of $10^{-4}$. On the other hand, the number of simulations is reduced by a factor of 10 at longer durations, i.e. a total of 1,000,000 and 100,000 simulation numbers are used for 5 to 50 years and 450 years, respectively.

## 2.2. Earthquake rupture modelling with space-time interaction

Ceferino et al. (2020) developed a space-time-dependent earthquake occurrence model, combining elements of elastic rebound theory with spatial rupture clustering at segment level in a relatively compact statistical formulation. In this model, the time between failure states (inter-arrival time) follows the BPT distribution, which arises from a Brownian Relaxation Oscillator (BRO) to represent the stress accumulation on faults (Matthews et al., 2002). The BRO resets the stress when it

reaches the failure threshold and thus reflects elastic rebound. Moreover, spatial stress interaction among neighbouring fault segments is modeled with a spatial correlation function represented by a spherical variogram model (Ceferino et al., 2020). Parameters are the mean ($\mu$) and coefficient of variation ($\alpha$) for each segment of the BPT distribution and the spatial decay rate ($\gamma$) for the spatial correlation of the segment ruptures (see section 2.2.2). Typically, the maximum likelihood estimation (MLE) is used to estimate rupture model parameters (Ogata, 1999; Zhuang et al., 2012), but the MLE is not robust when data

are scarce, as is the case in the Mentawai-Sunda zone. For instance, there were only two ruptures over 450 years in one particular segment (i.e. segment 6): the 1658 and 1797 events (see Figure 6A). We, therefore, use a Bayesian method




(Gelman et al., 2013; Ceferino et al., 2020) to estimate the model parameters. The following sections contain detailed descriptions of the procedures to model the earthquake rupture occurrence process.

### 2.2.1. Determination and segmentation of fault areas

We first define the source areas of the fault (see Figure 2A), focusing on the Mentawai region of the Sunda subduction zone. We collect supercyclic earthquake data of the Mentawai-Sunda subduction zone from the literature to define the tsunamigenic source geometry. Over the last 450 years, a total of 10 significant tsunamigenic events (> M 7.625) contributed to regional tsunami hazards (see Table 1). This number is consistent with the G-R model with at least one event (>M 7.625) every ~50 years (see Figure 1). The dates and spatial extents of the ruptures before the 19th century were inferred by
Philibosian et al. (2017) from paleogeodetic measurements in the Mentawai areas (see Figure 4). Specifically, coral microatoll samples from 21 sites along 600 km of the Mentawai region were used to constrain the dates, spatial extents and approximate earthquake source models (Natawidjaja et al., 2006; Shieh et al., 2008; Philibosian et al., 2014, 2017). Earthquake source models of recent tsunamigenic earthquakes (after 2000) were estimated from geodetic, seismic, and field measurements (Konca et al., 2008; Yue et al., 2014; see Figure 5). These finite-fault models contain areas of insignificant
slip along the edges (see Figure and Figure 5). Including these results in overestimated rupture areas (Mai and Beroza, 2002) and produces incorrect rupture lengths for the scenarios. We, therefore, exclude areas of insignificant slip by an effective dimension analysis. Further details about this analysis can be found in Mai and Beroza (2002) and Muhammad et al. (2016).

Using the above earthquake source models, we then define a one-dimensional (1D) representation of the fault from the effective along-strike lengths of the finite-fault models (see dark green color in Figure 6A). This 1D fault is then
discretized into six segments with a length of 100 km that corresponds to a minimum magnitude (M 7.75) of a significant tsunami-earthquake (see Figure 6B) and is consistent with the scaling relationship by Goda et al. (2016). This segmentation is an essential component of the space-time rupture model (see section 2.2.2). This minimum magnitude (i.e. M7.75) is the same as the central of the minimum magnitude for the PTHA of the time-independent model (i.e. see section 2.1.1).

To integrate the space-time rupture model and the stochastic source scenarios, we map simulated ruptures (Figure
6A and right panel of Figure 6B) onto discrete scenarios (left panel of Figure 6B) representing tsunami sources. Moreover, this setup is essential, allowing a good balance between the number of simulations and the spatial discretization of the fault area. A total of 21 source scenarios are developed to represent six magnitude levels (M 7.75, M 8.0, M 8.25, M 8.5, M 8.75, and M 9.0). The first six scenarios represent the minimum magnitude scenario (M 7.75). Moreover, the 7th to the 11th scenarios, the 12th to the 15th scenarios, the 16th to the 18th, and the 19th scenario are for the M 8.0, M 8.25, M 8.5, M 8.75,
and M 9.0 events, respectively (Figure 6B).

Each magnitude scenario is set up to a fixed length interval following the rupture length of the segment in the earthquake rupture modelling (see the length of each segment in Figure 6B). The maximum length of each magnitude is based on the total number of ruptured segments for each scenario. For example, the maximum length for the M 8.0 case is 200 km because two segments rupture in this scenario. When the ruptured length is more than half of the segment length





(e.g. >160 km for the M 8.0 case), the entire segment is involved in rupture simulations. The maximum and minimum

lengths of each magnitude scenario are listed in Table 2.

### 2.2.2. Space and time interaction approach

We adopt the space-time interaction approach by Ceferino et al. (2020) to model earthquake ruptures (Figure 7A). First, the

1D along-strike distance is discretized into 6 segments to represent the smallest areas that may rupture in tsunamigenic

earthquakes (see Figure 7A and section 2.2.1). Second, we set the rupture state of each segment $k$ at time $t$, $X_t(k)$, equal to 1

when a rupture occurs, and 0 otherwise. The rupture vector $X_t(k)$ depends on the rupture probability of each segment in year

$t$ conditioned on the time elapsed since the last earthquake, $X_t|T_t$, and is calculated according to a multivariate Bernoulli

distribution (Figure 7C):

$$X_t|T_t \sim Multivariate\ Bernoulli\ (p_{t,}\Sigma) \qquad (5)$$

where $T_t$ is the vector of elapsed time since the last rupture on each segment, $T_t = \{T_t(1), \dots, T_t(6)\}$ and is updated based

on the rupture occurrence ($X_t$). The relationship between $T_t$ and $X_t$ can be written as:

$$T_{t+1} = T_t(k)\{1 - X_t(k)\} + 1 \qquad . \qquad (6)$$

The two parameters $p_t$ and $\Sigma$ in Equation (5) describe the temporal and spatial interaction, respectively. $p_t$ is a

vector of the marginal probability of rupture on the $k$-th segment in year $t$ given the time since the last rupture ($T_t$). $\Sigma$ is a 6-

by-6 covariance matrix describing the spatial correlation of ruptures on the segments (see Figure 7C)

$p_t$ is calculated using the BPT cumulative distribution function and represents the rupture probability during a given year

conditional on no rupture since $T_t$ (Matthews et al., 2002; Zhuang et al., 2012; see left panel of Figure 7C):

$$p_{t(k)} = \frac{F(T_t(k)+1)-F(T_t(k))}{1-F(T_t(k))} \qquad (7)$$

where $F(t)$ is the BPT cumulative distribution function of inter-event waiting times defined by (Matthews et al., 2002):

$$F_k(t) = \Phi\left[\alpha_k^{-1}[t^{\frac{1}{2}}\mu_k^{-\frac{1}{2}} - t^{-\frac{1}{2}}\mu_k^{\frac{1}{2}}]\right] + e^{2/\alpha_k^2}\Phi[-(\alpha_k^{-1}[t^{\frac{1}{2}}\mu_k^{-\frac{1}{2}} + t^{-\frac{1}{2}}\mu_k^{\frac{1}{2}}])] \qquad (8)$$

We use a Bayesian approach to estimate the BPT parameters, i.e. mean ($\mu$), coefficient of variation ($\alpha$) and decay

rate ($\gamma$) (see Figure 7B). The Bayesian approach (Gelman et al., 2013) applied to a 450-year catalog produces the posterior

distribution of those parameters. A lognormal prior is adopted to carry out Bayesian simulation with the following

parameters:

1. The prior median of $\mu$ for each segment is different, namely 174, 50, 40, 45, 134, and 100 years for the segments 1 to 6,

respectively. These values represent the median interarrival time of earthquake rupture on each segment over the last

450 years. We set the prior logarithmic standard deviation of $\mu$ to 0.3 as it represents the deviation of the interarrival

time data.

2. We set the prior median of $\alpha$ to 0.7 in common with other time-dependent hazard assessments (Field et al., 2014;

Ceferino et al., 2020). We adopt a prior logarithmic standard deviation of 0.3.


On the other hand, the spatial correlation ($\rho$) among segments is represented by a spherical correlogram model (Ceferino et al., 2020). It defines the correlation of ruptures between segments $k$ and $l$ of the fault as a function of distance $D$ with a decay rate $\gamma$ (see right side of Figure 7C):

$$\rho_{k,l} = exp\left(-\left(\frac{D_{k,l}}{\gamma}\right)^2\right). \tag{9}$$

We estimate the decay rate $\gamma$ with the Bayesian approach along with the BPT parameters. We fix the prior median and logarithmic standard deviation of $\gamma$ to 400 km and 0.3, respectively, because the mean historical rupture length is about this size. The following equations (Equation (10)) show the formulae to estimate the posterior probability of the three parameters:

$$P(\mu_k, \alpha_k, \gamma | X) = \frac{P(X|\mu_k,\alpha_k,\gamma)P(\mu_k,\alpha_k,\gamma)}{\iiint P(X|\mu_k,\alpha_k,\gamma)P(\mu_k,\alpha_k,\gamma)d\mu_k d\alpha_k d\gamma} \tag{10}$$

$$P(\mu_k, \alpha_k, \gamma) = P(\gamma) \prod_{k=1}^{N} P(\mu_k)P(\alpha_k) \tag{11}$$

$$P(X_t|\mu_k, \alpha_k, \gamma) = \prod_{t=1}^{n_{years}} P(X_t|T_t) \tag{12}$$

$$P(X_t|T_t) = P(\cap_{k=1}^N A_k), where \begin{cases} A_k = \{Z_t(k) > \Phi^{-1}[p_t]\} \, if \, X_t = 0, \\ A_k = \{Z_t(k) \leq \Phi^{-1}[p_t]\} \, if \, X_t = 1 \end{cases} \tag{13}$$

To estimate the posterior, we first model the prior distribution using a multivariate lognormal distribution as presented in Equation (11) in each segment (i.e. $N = 6$). Second, we calculate the likelihood, $P(X_t|\mu_k,\alpha_k,\gamma)$, based on the
available rupture data (Equation (12)). It can be estimated as the product of the annual rupture occurrence probability distributions, $P(X_t|T_t)$, representing the probability of being in the intersection volume $A_k$, where $A_k$ is equal to either $Z_t(k) > \Phi^{-1}[p_t]$ when there is no rupture in year $t$ in the $i$th segment, and $Z_t(k) \leq \Phi^{-1}[p_t]$ otherwise (Equation (13)). Finally, we compute the denominator of the posterior with a Markov Chain Monte Carlo (MCMC) approach using the Metropolis Hasting (MH) algorithm (Chib and Greenberg, 1995).

**2.2.3.    Earthquake rupture simulation**

We use the estimated parameters to simulate earthquake ruptures using Equation (5). To obtain samples from the correlated multivariate Bernoulli distribution (Equation (5)), which cannot be written in a closed form, Ceferino et al. (2020) adopted a copula method (Jin et al., 2015) to obtain approximate samples (Equation (14)). Figure 7D illustrates the procedure to sample rupture occurrence of each segment with the Gaussian copula. First, we generate correlated normally distributed
random variables ($\mathbf{Z_t}$) with zero mean and a covariance matrix $\Sigma$ defined by the correlogram model (Equation (9)) using



Equation (14). The matrix size of the correlogram model is based on the total number of segments, i.e. the size of the matrix is $N \times N$ where $N = 6$ (i.e. a total number of segments):

$$\boldsymbol{Z_t} \sim \mathcal{N}(0, \Sigma_{k,l}), \text{ where } \Sigma_{k,l} = \rho_{k,l}. \tag{14}$$

We then transform $Z_t$ using the standard normal cumulative distribution function $\Phi(Z_t)$ to obtain correlated, uniformly distributed random variables in [0,1]. $X_t$ is set to 1 (rupture has occurred) when $Z_t$ is smaller than $p_t$ from Equation (7) and set to 0 otherwise:

$$X_{t(k)} = \mathbb{1}\{\Phi(Z_t(k)) < p_t(k)\}. \tag{15}$$

The bottom panel of Figure 7D illustrates a synthetic rupture catalog simulated with the copula approach by sampling the correlated multivariate Bernoulli distribution. The profile of ruptures at the 20th year of simulation (highlighted by the red line in Figure 7D shows that segments 4, 5, and 6 have ruptured because their $\Phi(Z_t)$ values are both less than $p_t$, whereas other segments do not rupture.

Finally, we carry out Monte Carlo earthquake rupture simulations. First, the number of t-years earthquake catalogs is set to $N_{sim} \geq 100{,}000$ (see section 2.1.2). Second, using the spatial correlation among the segments ($\rho_{k,l}$) (Equation (9)) and the initial elapsed time since the last past earthquake ($T_t$), we simulate correlated rupture probabilities in one-year increments with the copula method. We then determine rupture occurrences in year $t$ ($\boldsymbol{X_t}$) using (Equation (15)) in each segment. We finally repeat the procedure to obtain ≥100,000 catalogs over the periods of interest.

### 2.3. Stochastic earthquake source generation and tsunami simulation

We estimate tsunami hazard in the region of interest via a Monte Carlo tsunami simulation (see Figure 2C) by generating stochastic earthquake source models. To generate source models, we first sample earthquake source parameters, including rupture width ($W$), mean slip ($D_a$), maximum slip ($D_m$), Box-Cox parameter ($\lambda$), correlation length along strike direction ($A_x$), correlation length along dip direction ($A_z$), and Hurst number ($H$). We use the empirical scaling relationships for global subduction earthquakes derived by Goda et al. (2016) because they are consistent with the source parameters of the past tsunamigenic events in the Sunda subduction zone (Muhammad et al., 2016).

Among these source parameters, the rupture length ($L$) is pre-defined at the beginning of the simulation to integrate the tsunami hazard assessment and the earthquake rupture modelling. The length is calculated based on the magnitude and fault segmentation (see section 2.2.1). Therefore, given a pre-defined fault length, we randomly draw the other source parameters from the source scaling relationships. To ensure the consistency of the generated source parameters using the scaling relationships, we require that the simulated seismic moment ($M_o = \mu W L D_a$, where $\mu$ is the rock rigidity) is within a tolerance of plus/minus 0.1 magnitude units of the target moment of a magnitude scenario. We assume a rigidity of 40 GPa, which is typical for the Sunda subduction zone (Natawidjaja et al., 2006; Kongko et al., 2011). As a result, we produce





simulated values of W and Da iteratively until the seismic moment condition is met (i.e. *L* has been pre-defined at the beginning of the simulation).

We next develop stochastic source models. We first generate a random field using the Fourier integral method (Mai and Beroza, 2002) based on the simulated spatial slip distribution parameters. We use a Box–Cox transformation to obtain slip distributions with realistic positive skewness. We adjust the transformed slip distribution to achieve the target mean slip, $D_a$, and avoid very large slip values exceeding the maximum target slip, $D_m$. Muhammad et al. (2016, 2017) provide further details about this method. Based on a stability test of the tsunami simulations, 300 stochastic models are sufficient to simulate stable and consistent tsunami heights and depths (De Risi and Goda, 2017). We generate 300 source models for each scenario for a total of 6,300 stochastic source slip models.

Given the source models, we simulate tsunamis to estimate the tsunami hazard along the western coast of Sumatra, which requires a digital elevation model (DEM) and bathymetry data. Bathymetry data are taken from the GEBCO2014 dataset (http://www.gebco.net/data_and_products/gridded_bathymetry_data/), whilst SRTM1 (https://lta.cr.usgs.gov/SRTM1Arc) is used for the DEM data (see Figure 8A). For areas around the target zone of Pariaman, Padang, and Painan, we adopt local DEM (DEM5) and Bathymetry (Bathy5) within a nested grid system of four resolution levels. Therefore, the tsunami simulation grid for Sumatra are 1,350 m (red box in Figure 8B), 450 m (green box in Figure 8B), 150 m (magenta box in Figure 8B), and 50 m (blue box in Figure 8B). The 1,350 m region encompasses all of Western Sumatra.

Once stochastic source models are generated, and the DEM/bathymetry data are constructed, we calculate the initial water surface elevation using formulae by Okada (1985) and Tanioka and Satake (1996) that consider the deformation due to both vertical and horizontal displacements of the seafloor. Tsunami waves are then propagated by solving the nonlinear shallow water equation with runup (Goto et al., 1997). The effects of surface roughness on tsunami flows are modelled with Manning's bottom friction formula with a uniform roughness coefficient of 0.025 $m^{-1/3}$s for the ocean and 0.06 $m^{-1/3}$s for land (Griffin et al., 2016). The fault rupture is supposed to happen instantly, whilst the tsunami simulation runs for 2 hours with a time step of 0.5 seconds to meet the Courant-Friedrichs-Lewys requirement for bathymetry and elevation data for the Mentawai region (Selvan and Kankara, 2016; Akoh et al., 2017). The above simulation procedure is run iteratively through a Monte Carlo simulation for all stochastic source models. Subsequently, tsunami simulation results at locations of interest are evaluated. Furthermore, the Monte Carlo tsunami simulation results can be used to assess the variability of tsunami simulation outcomes in terms of tsunami wave height at different locations (e.g. Pariaman, Padang, and Painan; see Figure 8C) and for developing probabilistic tsunami hazard assessments.



## 3. Results and Discussion

### 3.1. Bayesian Parameter Estimation

Using the prior parameters (see section 2.2.2), the MCMC algorithm sampled 10,000 posterior models. The chain explores a 13-dimensional parameter space: six pairs of $\mu$ and $\alpha$, and one $\gamma$. The standard deviation of random walk steps for all $\mu$ and $\alpha$ and $\gamma$ are 12.5, 0.1, and 25, respectively. These parameters produce an acceptance rate of ~20%, indicating good mixing (Chib and Greenberg, 1995; Sherlock and Roberts, 2009). We observe stable parameter estimates after a burn-in period of 500 steps.

Figure 9 illustrates the MCMC results with the priors and posteriors of $\mu$, $\alpha$, and $\gamma$ for all segments, whilst the final parameter estimates are taken from the maximum a posteriori (MAP; Table 3). The figure shows that, in general, the available earthquake data can effectively reduce the parametric uncertainty of the priors, in particular for the $\mu$ parameter in segments 2 to 4. The median interarrival times of the central segments (i.e. segments 2 to 4) are about 40 years, while the interarrival times in the remaining segments are more than 50% greater. The uncertainties of the parameters for the interarrival times of segments 1, 5, and 6 are large because few ruptures have occurred in those segments (see Figure 6). Moreover, the data dispersion of the central segments is greater than in others, resulting in a higher coefficient of variation. On the other hand, the estimated $\gamma$ is about 500 km, indicating substantial spatial correlation of ruptures. In general, the estimated parameters reflect the interarrival times and variations of the earthquake data (see Table 1 and Figure 6).

### 3.2. Earthquake rupture modelling

We model earthquake ruptures with the Bayesian parameter estimates (see section 2.2). To simulate earthquake ruptures over the seven considered periods, i.e. 1, 5, 10, 20, 30, 50, and 450 years, we first generate ruptures for the longest period (450 years) by considering a total of 100,000 catalogs. We then simulate ruptures for the 50 years with a total of 1,000,000 catalogs and further construct the other period catalogs (i.e. 5, 10, 20, and 30) based on the 50-year simulations. Finally, the 1-year rupture simulation is performed with a total of 10,000,000 catalogs. To validate the rupture simulations, we further compare the average annual seismic moment release simulated over 450 years with the observed moment release. The seismic moment, $M_0$, is calculated for each segment according to (Hanks and Kanamori, 1979) in units of $10^{-7} Nm$:

$$M = \frac{2}{3}\log(M_0) - 10.7. \tag{16}$$

The calculated seismic moments for both simulations and observations of each event are distributed uniformly over each ruptured segment. We then sum $M_0$ within each segment and divided by 450 years. Figure 10 compares the average annual moment release between the 450-year simulation and the observed historical data. The figure shows that the simulations are consistent with the data and hence, reasonably approximate rupture catalogs for the time-dependent PTHA.

We then present an example of one simulated catalog (number 152) for the next 5, 50, and 450 years in Figure 11. The figure shows that for the short periods ($\leq 50$ years), only a few earthquakes occur in the central segments 2 to 4 (Figure





11A-B). The number of events increases significantly after 50 years (see Figure 11C). Moreover, to understand which scenarios are produced from the earthquake rupture simulation (see Figure 6B), Figure 12 provides a relative frequency of each scenario from the 100,000 catalogs of the seven periods, including 1, 5, 10, 20, 30, 50, and 450 years. In general, over the next 5 years, the chances of segments 1 to 4 and segments 2 to 4 rupturing simultaneously are relatively small. The

relative frequency of scenario 16 (i.e. segments 1 to 4; see Figure 6B) is 0.0037%, whereas the summed relative frequency of scenarios 12 (i.e. segments 1 to 3), 13 (i.e. segments 2 to 4), and 16 is about 19%. Such a trend reflects the conditioning on past earthquake rupture history, i.e. in this context, segments 1 to 4 ruptured in 2007/2010, so the chances are small that another large one will happen here soon.

In general, the results (Figure 11C) show that segments 1, 5, and 6 have fewer ruptures due to greater recurrence

intervals (i.e. ~120 years to ~180 years) than segments 2, 3, and 4 with shorter interarrival times (i.e. ~40 years). Consequently, scenarios 8 and 13 occur the most, followed by scenario 9. To produce significant tsunamigenic events (>M 8.75), at least five segments need to rupture. Subsequently, the relative frequency of these significant magnitudes (>M 8.75) is small (below ~5% for the period of over 10 years). In addition, the number of significant tsunamigenic earthquakes greater than M 8.25 increases with longer periods (i.e. 450 years). The simulations also suggest that the number of an M 9.0 event

over the next 450 years is non-zero (see Figure 12O). Longer periods of simulation (≥ 50 years) approach the magnitude distribution presumed by the time-independent PTHA method. However, it is noted that the distributions of the time-independent and time-dependent models are not identical: the time-independent case is based on the G-R assumption, whilst the time-dependent approach includes the effects of fault rupture interaction.

### 3.3. Time-dependent PTHA

This section presents the main time-dependent PTHA results, using the results of the rupture modelling and the stochastic tsunami simulations (see section 2.1.2). We first illustrate the development of a tsunami hazard curve representing the time-independent and the time-dependent exceedance probability of tsunami height. Second, we compare time-dependent and time-independent PTHA. Finally, we present and discuss the PTHA for the entire western coast of Sumatra.

### 3.3.1. Development of the tsunami hazard curve

We illustrate the hazard curve development (section 2.1) at a location in Pariaman City (P1 in Figure 8C). In the time-independent PTHA, the simulated tsunami heights of each magnitude/source are used to construct the empirical CCDF. The conditional hazard curve for each magnitude is further calculated as the product of the CCDF, and the probabilities of each magnitude (i.e. the probabilities of M 7.75, M 8.0, M 8.25, M 8.5, M 8.75, and M 9.0 are 0.47, 0.26, 0.14, 0.076, 0.042, and 0.023, respectively; see Figure 1D). Subsequently, the mean annual rate, $\lambda(H \geq h)$, is calculated as a sum of the conditional

hazard of each magnitude multiplied by the mean annual rate $\lambda(M \geq 7.625)$, which is equal to 0.034. The mean annual rate is estimated from the instrumental catalog of the Mentawai-Sunda zone. The mean annual rate parameter is finally used to calculate the probability $P(H \geq h|t)$ of exceeding a specific $h$ for a given $t$ years.





Figure 13 visualizes the development of hazard curves for Pariaman City at the depth of about 0 m (P1 in Figure 8C) using the time-independent PTHA. First, all simulated heights from the same magnitude are combined and the CCDFs of those six

magnitudes are calculated (see Figure 13A). The conditioned hazard curve of each magnitude and the final unconditional hazard curve are further generated and presented in Figure 13B and Figure 13C, respectively. Adopting the annual hazard rate in Figure 13C, a hazard curve for a given time window (i.e. 1, 5, 10, 20, 30, 50 and 450 years), $P(H \geq h|t)$, is then calculated using Equation (1) as presented in Figure 13D.

Figure 14 displays the development of hazard curves for the time-dependent PTHA at a location in Pariaman City

(P4 in Figure 8C) generated from one catalog of ruptures over 50 years (again, catalog 152). Figure 14B presents one synthetic 50-year rupture catalog in which three tsunamigenic scenarios occurred (the 4[th], 8[th], and 15[th] scenarios, Figure 14A). The heights from the stochastic tsunami simulation are further taken from the corresponding scenarios and are presented in Figure 14C. One height for each scenario is then randomly chosen from the 300 simulated heights of each occurred scenario. Those sampled heights are further used to define the maximum height for each catalog (i.e. one height for

one simulation catalog). Such procedures are then repeated for each rupture catalogue to produe a total of $N_{sim}$ maximum heights in a specific $t$-year simulation ($N_{sim}$ for the 50-year simulation is 1,000,000 catalogs). Using these maximum heights, the final hazard curve is finally developed following the procedure in section 2.1.2 and is presented in Figure 14D.

### 3.3.2.    Comparison of time-dependent and time-independent PTHA

We first illustrate the differences between the time-dependent and time-independent PTHA in terms of hazard curves at three

cities along the western coast of Sumatra, including P4 in Pariaman, P12 in Padang, and P25 in Painan located at depths between 0.5 and 1.5 m (see Figure 8C). We then construct tsunami hazard maps from the time-independent, and time-dependent PTHA approaches for comparison. Figure 15 shows a comparison of tsunami hazard curves in terms of $P(H \geq h|t)$ based on the time-dependent and time-independent PTHA (see magenta line in Figure 15) for 1, 5, 10, 30, 50, and 450 years.

For the 1-year case, the time-dependent PTHA generally produced a higher hazard probability than the time-independent PTHA model, specifically at the tsunami height of ≤ 5 m. At the tsunami height of 2 m, where it is relatively hard to evacuate (Pregnolato et al., 2017), the time-dependent model is about 20-25% higher than the time-independent model. This is due to the frequent earthquake ruptures in the central segments (segments 2 to 4) found in 5% of the 10,000,000 catalogs. Past ruptures in the central segments have observed recurrence intervals between 3 (i.e. occurred

between 2007 to 2010) and 20 years (i.e. occurred between 1597 to 1613). Hence, the ruptures in the 1-year simulation are located mainly in the central segments of the Mentawai-Sunda zone. This trend persists along the coast of western Sumatra (Figure 16). This figure presents the mean hazard curves at all points along the western coast of Sumatra located at a depth between 0.5 and 3 m, from P1 to P33 (see Figure 8C). The figure confirms that the time-dependent probability of a tsunami height below 4-5 m is, in general, higher than the time-independent model at all points.





In addition, similar differences between the time-independent and the time-dependent models persist over shorter periods, specifically for ≤20 years (see Figure 15A-L). In contrast, the time-dependent probabilities of heights above 5 m over short periods are smaller than the time-independent model because almost no M 9.0 events occur during these periods (see Figure 12A-L). Moreover, starting from the period of 30 years, the time-independent mean hazard probabilities are ~10-30% greater than the time-dependent model, particularly at the tsunami level of >5 m for 30 and 50 years, respectively (see

Figure 15M-R). However, the differences decrease with a longer period.

     Over the 450-year simulation, the time-dependent and time-independent models generate similar hazard probabilities, particularly for heights less than 5 m. However, differences remain noticeable above 5 m. The following reasons can explain the differences. The chances of M 8.75 and M 9.0 events producing significant tsunami heights (>3 m) in the time-dependent model are very slim. This is indicated by the small relative frequencies of only 3.2% and 0.61% for the

M 8.75 and M 9.0 events, respectively. In contrast, the time-independent model prescribes a probability of 4.2% and 2.3% for the M 8.75 and M 9.0 events and hence, these two events may contribute more to the higher hazard probability values than in the time-dependent model. The small chance of having the M 9.0 events for the time-dependent model is also because the segments from the maximum past tsunamigenic events (i.e. the M 8.8 1797 and the M 8.9 1833 events) ruptured only four segments (see Figure 2A). Therefore, adopting this past characterization leads to a small number of future events

with a magnitude of above M 8.75.

     Moreover, the difference in hazard curves over the long period is due to the different approaches in modelling the recurrence of the tsunamigenic event. The time-dependent model adopted the temporal renewal processes considering the time since the last earthquake. In contrast, the time since the last earthquake does not feature in the time-independent model. With a recurrence time of a significant tsunamigenic event (> M 8.75) of ~ 250 years (see Figure 2A), the failure state of the

time-dependent rupture model is reset on average every 250 years and hence, even for a period longer than 450 years, the hazard probability of the time-independent model will still exceed the time-dependent model.

     Regionally, our results show that Painan City experiences greater tsunami hazards than the other two cities. The tsunami heights at the coast of Painan City can be as much as 15 m compared to ~10-12 m in Pariaman and Padang (see Figure 15). Segments 2 to 4 of the Mentawai region have a greater rupture probability and are located close to Painan City, hence driving the tsunami hazard level higher than in the other two cities. However, the tsunami hazard levels in Pariaman

and Padang are still significant, as shown by high probabilities (~ 40%) of tsunami heights greater than 2 m. The tsunami evacuation would face major difficulty at this 2-m level of tsunami height (Pregnolato et al., 2017), leading to a significant number of fatalities (> 300 people; Muhammad et al., 2021). Therefore, a well-prepared tsunami mitigation system is extremely important along the western coast of Sumatra.

Differences over shorter periods are further exhibited in 50-year tsunami hazard maps. In general, 50 years are used to represent the hazard maps at different probability levels for hazard assessment purposes (Akinci, 2009; De Risi and Goda, 2017). For the time-independent approach, maps for different probabilities in 50 years may represent various return periods, e.g. 10% and 2% in 50 years correspond to return periods of 475 years and 2475 years, respectively. Figure 17 shows the




tsunami hazard maps in Padang City based on the time-dependent and time-independent methods for two different
probabilities (10% and 2%) in 50 years. The maps are only developed for Padang City because a high-resolution (~5 m) of
DEM and bathymetry is used to construct the tsunami simulation dataset, producing a realistic inundation level over land. In
Figure 17, depth indicates the hazard level in Padang. The figure shows that the mean hazard levels from both time-
dependent and independent models are similar, with the hazard level below 5 m for the 50 years simulation at a 2%
probability (see Figure 15P-R). Thus, the maps show a similar hazard level between the time-dependent and the time-
independent models.

## 4. Conclusions

This study developed a time-dependent probabilistic tsunami hazard analysis for Western Sumatra. A new framework of the
time-dependent PTHA adopting space and time interactions of earthquake ruptures was used to simulate future earthquake
ruptures in the Sunda subduction zone. The Mentawai region of the Sunda subduction zone was considered as a case study to
develop the tsunamigenic sources. Three important cities, namely Pariaman, Padang and Painan, were chosen as localities to
assess the tsunami hazard using the time-dependent PTHA method. Another innovation in this study was to integrate
stochastic tsunami simulation and the space-time interacting earthquake rupture modelling to develop time-dependent PTHA
and compare it with a time-independent approach. The tsunamigenic source used to run the stochastic tsunami simulation
adopts the earthquake supercyclic rupture areas of the Mentawai-Sunda zone in the last 450 years. Six magnitude scenarios,
M 7.75, M 8.0, M 8.25, M 8.5, M 8.75, and M 9.0, were considered for running the stochastic tsunami simulation. A total of
21 tsunami scenarios from 6 different magnitudes were adopted to generate the stochastic source models and then used to
run the Monte Carlo tsunami simulations. Moreover, in the space-time interaction model, the BPT distribution and the spatial
correlogram model were introduced to generate the time between failure states (inter-arrival time) and the stress interaction
among the neighbouring zones within the tectonic region, respectively. To determine parameters in developing such models,
490 a Bayesian MCMC simulation was adopted. Subsequently, the final hazard curve and the hazard maps from the time-
dependent PTHA were obtained using the results from the stochastic tsunami simulation and the earthquake rupture
modelling.

The earthquake rupture modelling results showed that the calibrated space-time interaction model successfully
generates annual seismic moment release rates consistent with observations over the last 450 years. The time-dependent
495 model also captured the uncertainty of future tsunamigenic events by producing a wide range of earthquake rupture catalogs.
Comparing the mean hazard probabilities between the time-dependent and the time-independent approach indicated that the
two approaches produce similar results. However, the time-dependent model produces 10-25% higher hazard probability at a
height level of ≤ 5 m for the short periods (≤ 20 years). Therefore, the time-independent approach may underestimate the
hazard level for a short-term period and hence, we recommend considering the time-dependent model for short-term hazard
500 assessment. In contrast, the time-independent model developed for a long-term (> 30 years) hazard assessment produced
much higher hazard probabilities, by up to ~30%, than the time-dependent model. The time-independent model may be
suitable to capture the worst-case scenario for future tsunami hazard and risk assessment analysis.





Regionally, the PTHA assessment along the western coast of Sumatra showed that the southwestern part of Sumatra (Painan) located closer to the Mentawai-Sunda zone may experience more significant tsunami damage than the northwestern part (Padang and Pariaman). However, the tsunami hazards were still significant in those three cities, shown by >5 m of tsunami height at the 50 years of simulation. The time-dependent PTHA approach may also produce more accurate tsunami hazard assessment results for short-term periods (≤ 50 years). Therefore, it is essential to adopt the time-dependent PTHA to update the tsunami mitigation plans regionally because the Mentawai-Sunda zone is expected to have significant tsunamigenic events in the next few decades and currently, there has not been time-dependent PTHA work carried out there.

## Acknowledgements

The authors gratefully acknowledge financial support from the University of Bristol through the 2017 Strategic Research Fund for the project "Modelling Seismic Risk Cascades: Towards Time-Dependent Earthquake Impact Assessment". This work was also supported by the Leverhulme Trust (RPG-2017-006) for the Global Earthquake Resilience for Natural-Engineering-Social Interacting Systems (GENESIS) project. M. J. W. received funding from the European Union's Horizon 2020 research and innovation program under Grant Agreement Number 821115, Real-Time Earthquake Risk Reduction for a Resilient Europe (RISE). The bathymetry and elevation data for the Sumatra region were obtained from the GEBCO2014 database (http://www.gebco.net/data_and_products/gridded_bathymetry_data/) and the SRTM1 database (https://lta.cr.usgs.gov/SRTM1Arc), respectively.

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





**Table 1 Past significant tsunamigenic events in the Mentawai-Sunda subduction zone over the last 450 years**

| No. | Year | Magnitude (M) | Recorded tsunami height | References |
|---|---|---|---|---|
| 1 | 1597 | 8.4 | N/A | Sieh et al. (2008); Philibosian et al. (2017) |
| 2 | 1613 | 8.4 | N/A | Sieh et al. (2008); Philibosian et al. (2017) |
| 3 | 1631 | 8.3 | N/A | Sieh et al. (2008); Philibosian et al. (2017) |
| 4 | 1658 | 8.4 | N/A | Sieh et al. (2008); Philibosian et al. (2017) |
| 5 | 1703 | 8.6 | N/A | Sieh et al. (2008); Philibosian et al. (2017) |
| 6 | 1797 | 8.8 | 5-7 m in Padang and Bengkulu | Natawaidjaja et al. (2006); Shieh et al. (2008); Philibosian et al. (2014, 2017) |
| 7 | 1833 | 8.9 | 5-7 m in Padang, Bengkulu and Manna | Natawaidjaja et al. (2006, 2007); Shieh et al. (2008); Philibosian et al. (2014, 2017) |
| 8 | 2007a | 8.4 | No significant tsunami | Ji (2007); Konca et al. (2008); Gusman et al. (2010) |
| 9 | 2007b | 7.9 | No significant tsunami | Ji and Zheng (2007); Konca et al. (2008) |
| 10 | 2010 | 7.8 | 3-5 m in the coastal areas of Pagai Islands | Hayes et al. (2012); Satake et al. (2013); Yue et al. (2014) |

**Table 2. Maximum and minimum lengths of each magnitude scenario**

| | Fault length (km) | |
|---|---|---|
| Magnitude | Minimum | Maximum |
| | m | m |
| 7.75 | 60 | 100 |
| 8.00 | 160 | 200 |
| 8.25 | 260 | 300 |
| 8.50 | 360 | 400 |
| 8.75 | 460 | 500 |
| 9.00 | 560 | 600 |




**Table 3. Bayesian parameter estimates of marginal BPT distributions of ruptures and spatial correlation.**

| Segment | Maximum a posteriori (MAP) | | |
|---|---|---|---|
| | $\mu$ (years) | $\alpha$ | $\gamma$ (km) |
| 1 | 173 | 0.62 | |
| 2 | 36 | 1.05 | |
| 3 | 36 | 1.16 | 500 |
| 4 | 44 | 0.96 | |
| 5 | 121 | 0.62 | |
| 6 | 147 | 0.66 | |


**Figure 1. (A) Earthquake epicentres in the Mentawai-Sunda zone from 1970 to 2015 with magnitudes > 5.0. (B) *b* value estimation.**
**(C) Mean annual rate of earthquake occurrence. (D) Discretized probability distribution of magnitude.**




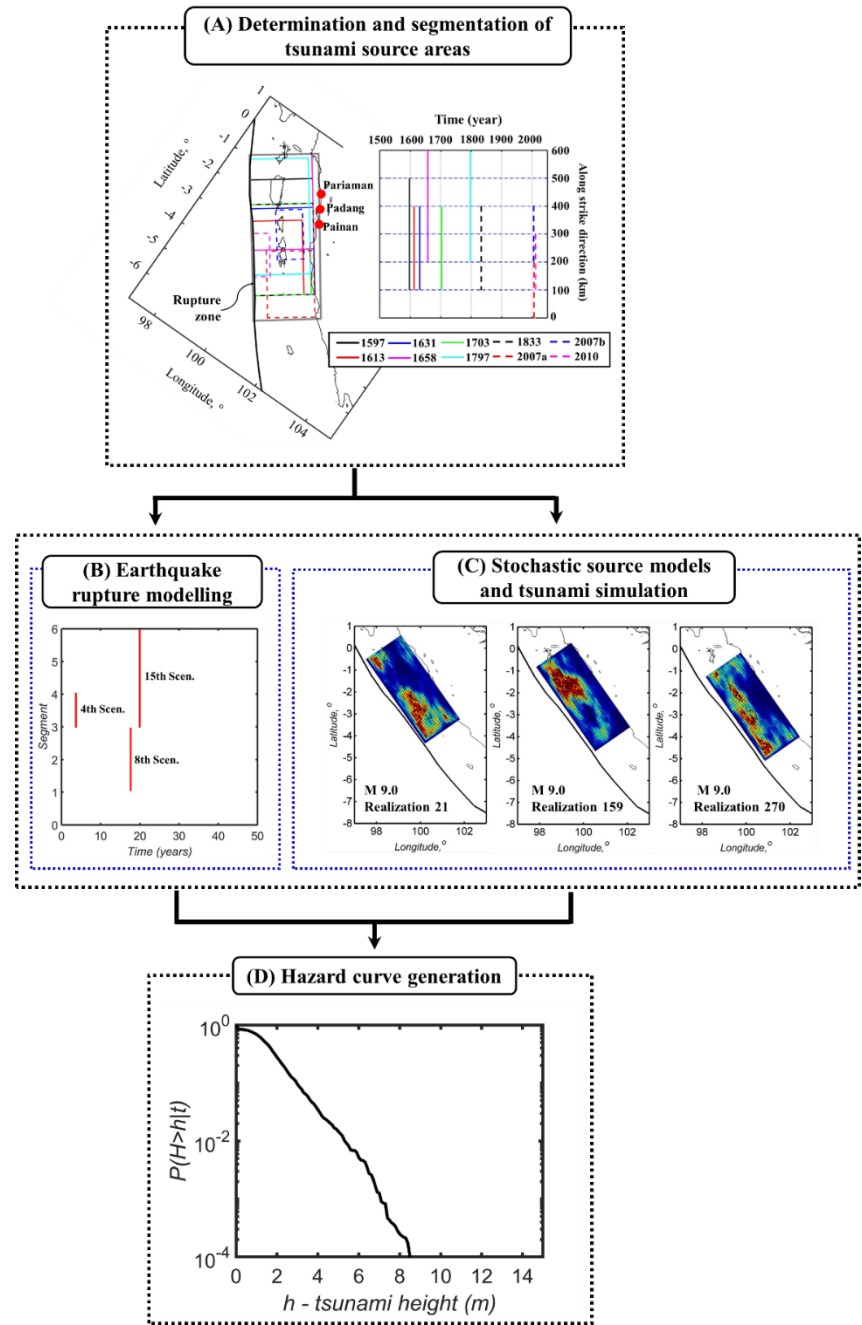

**Figure 2. Time-dependent PTHA framework. (A) Determination and segmentation of tsunami source areas based on the supercyclic tsunamigenic earthquakes in the Mentawai-Sunda zone. (B) An example of one simulation of earthquake rupture modelling. The 4th, 8th, and 15th scenarios refer to different rupture scenarios. (C) Stochastic tsunami source simulation (stochastic earthquake slip models). (D) Generation of hazard curves.**




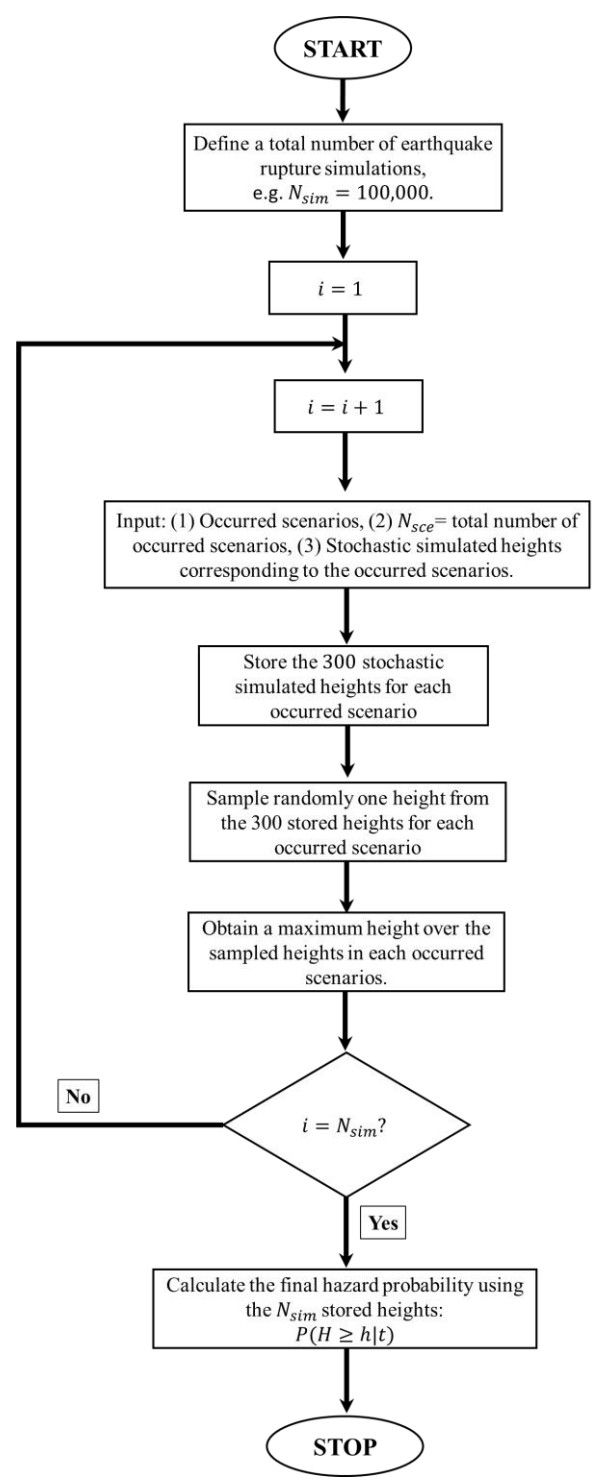

**Figure 3 Procedure to generate the time-dependent mean hazard curve.**


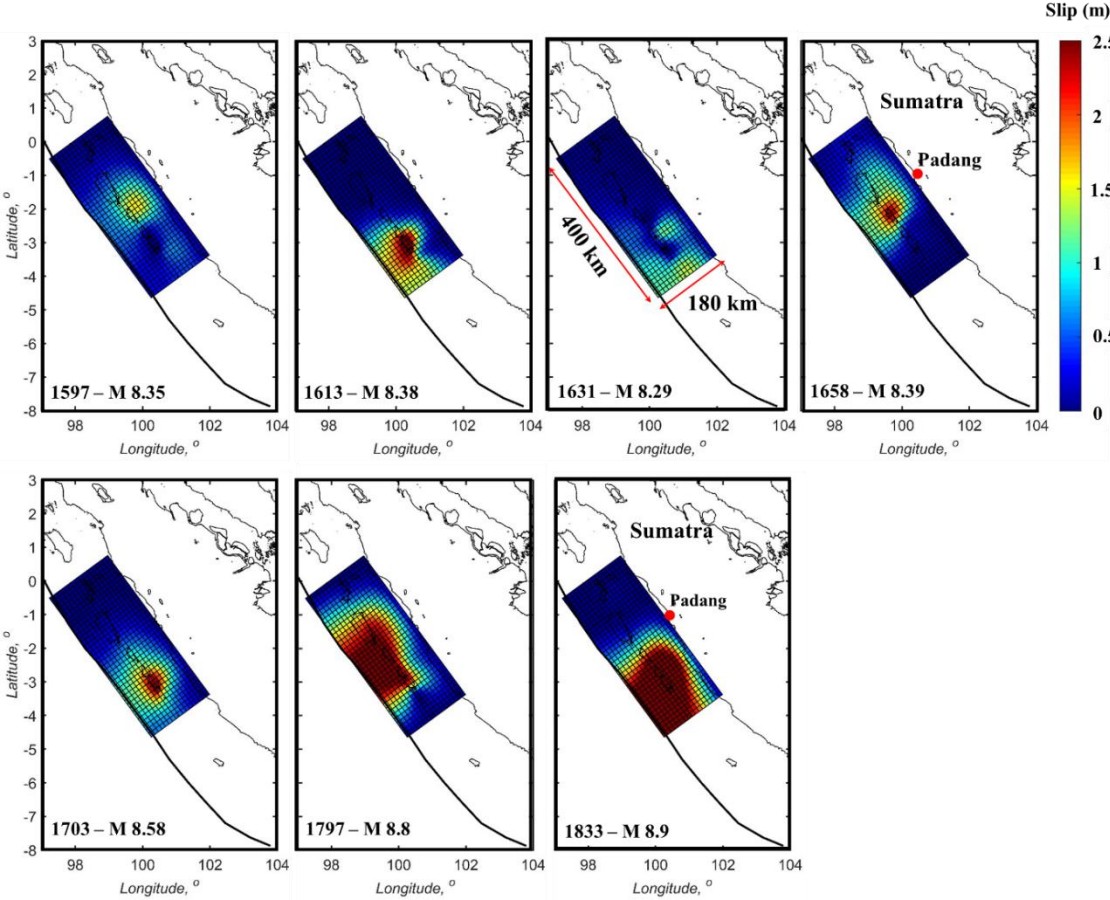

**Figure 4. Source models of tsunamigenic earthquakes between the 16ᵗʰ and 19ᵗʰ centuries inverted from paleogeodetic measurements by Philibosian et al. (2017).**




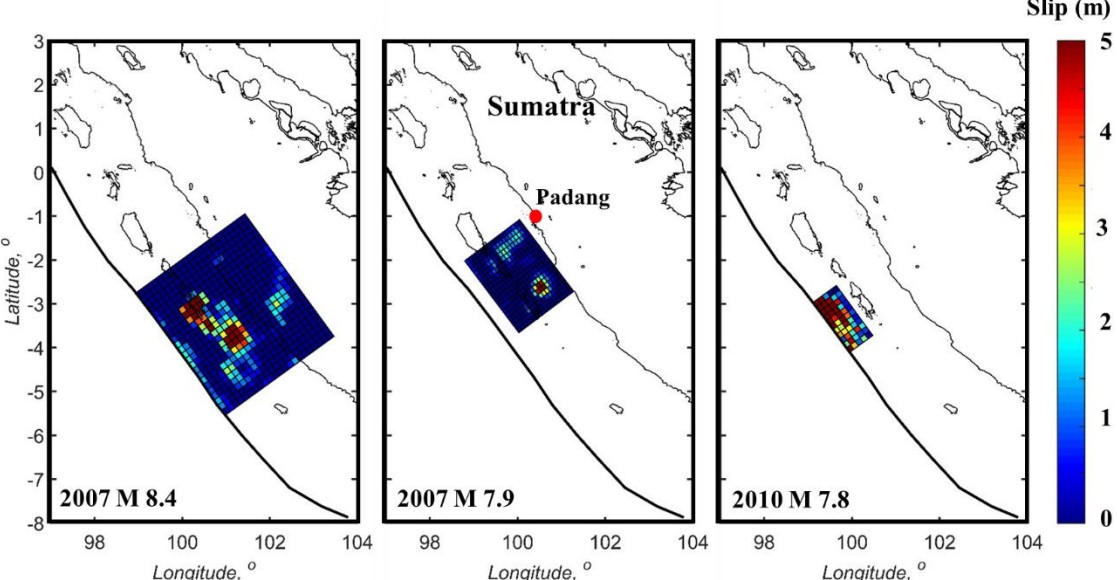

**Figure 5. Source models of recent earthquakes that are constrained by geodetic measurements and seismological data (Konca et al., 2008; Yue et al., 2014).**
**Figure 6. (A) One-dimensional (1D) representation of the fault from effective along-strike lengths of the earthquake source models from past tsunamigenic events in the Mentawai-Sunda zone. It covers the three study areas: Pariaman, Padang, and Painan. (B) Discretization of tsunamigenic sources.**




**Figure 7. Procedures for modelling spatially-correlated earthquake ruptures on segments with marginal Brownian Passage Time (BPT) probability distributions.**


**Figure 8. (A). Digital Elevation Model and bathymetry for the Mentawai-Sunda region (GEBCO, 2017). (B) Nested grids used for tsunami simulation in Padang. (C) Detailed study areas where the simulated tsunami heights are computed (Source of base map: © Google Maps).**







**Figure 9.** Bayesian parameter estimation of *μ*, *α*, and *γ* for all segments. The posterior histogram is calculated from the simulation, whilst the blue and the red lines represent the PDF of the prior and the estimated posterior.


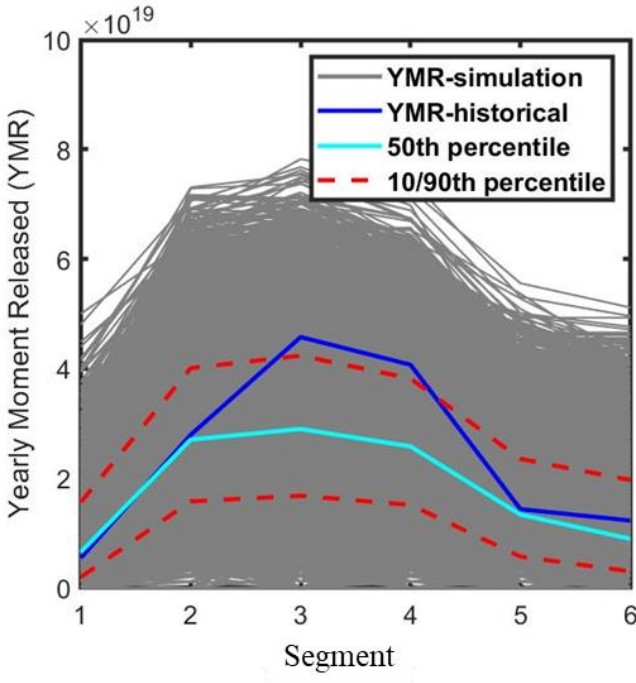

**Figure 10. Comparison of observed and simulated annual seismic moment release. The red lines represent the 10th and the 90th percentile of the simulated annual moment released, whilst the cyan color shows the 50th percentile of the simulation annual moment released.**




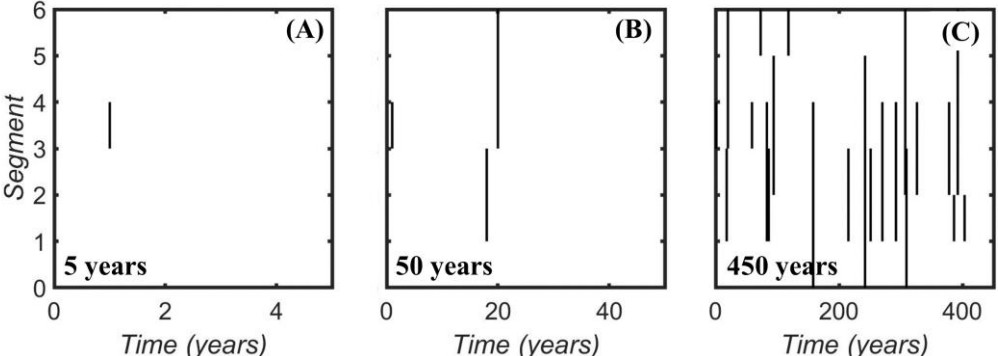

**Figure 11. Results of earthquake rupture simulations for three different periods, i.e. 5 years (A), 50 years (B), and 450 years (C) from one simulation catalog (number 152).**




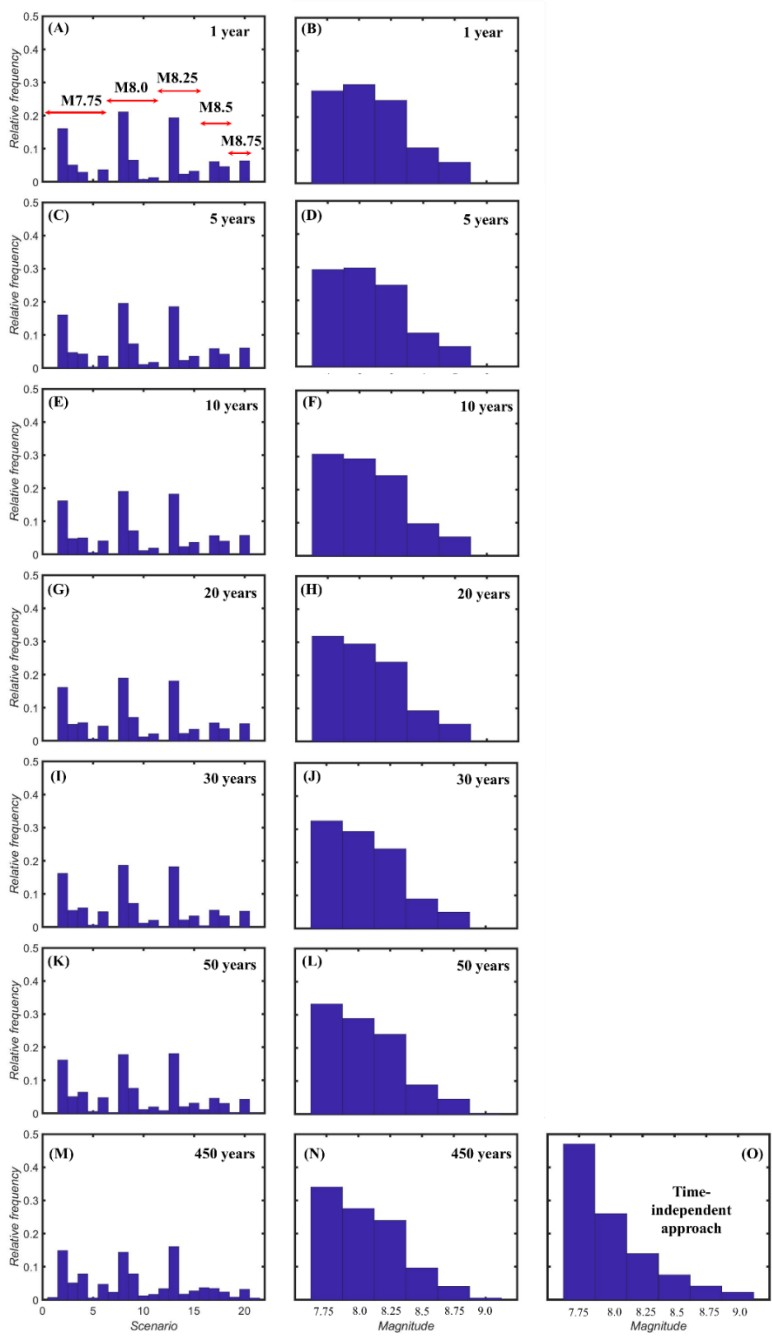

**Figure 12. Percentage of occurrence of each scenario (left panel) and magnitude (central panel) produced from the rupture**
**simulation over seven periods: 1 year (A and B), 5 years (C and D), 10 years (E and F), 20 years (G and H), 30 years (I and J), 50**
**years (K and L), and 450 years (M and N). For comparison, the probability values of each magnitude from the time-independent**
**approach are plotted in (O).**


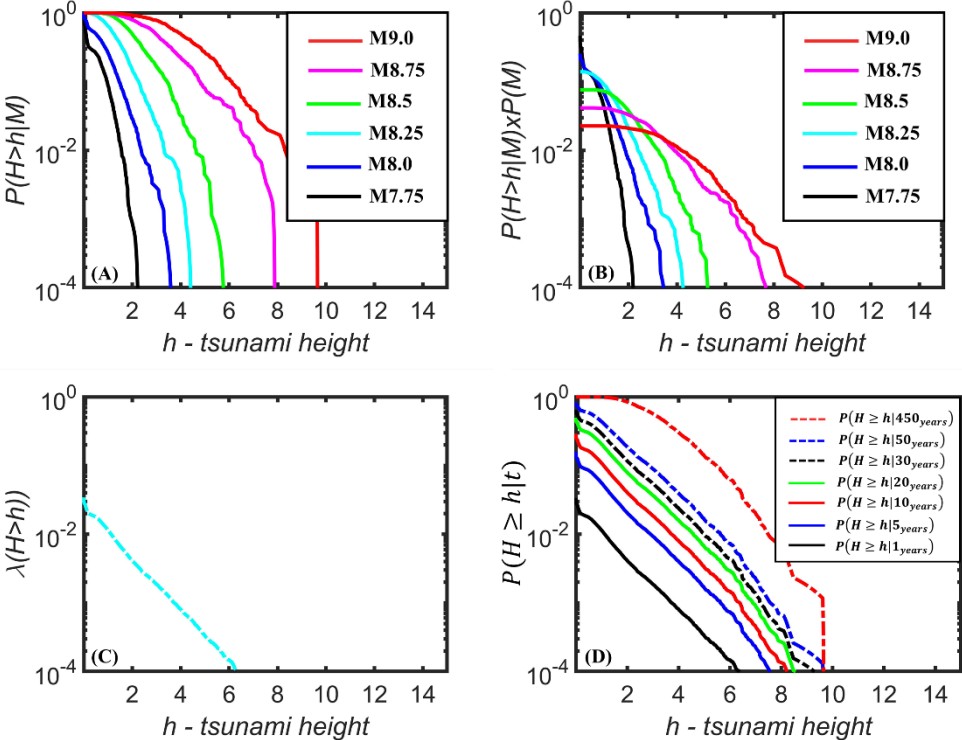

**Figure 13. Development of final hazard curve for the time-independent model: (A) conditional hazard curve of each magnitude, (B) the conditioned hazard curve weighted by the probability values obtained from the discretized Gutenberg-Richter relationship, (C) mean annual hazard rate, and (D) the final hazard curve conditioned to considered periods.**



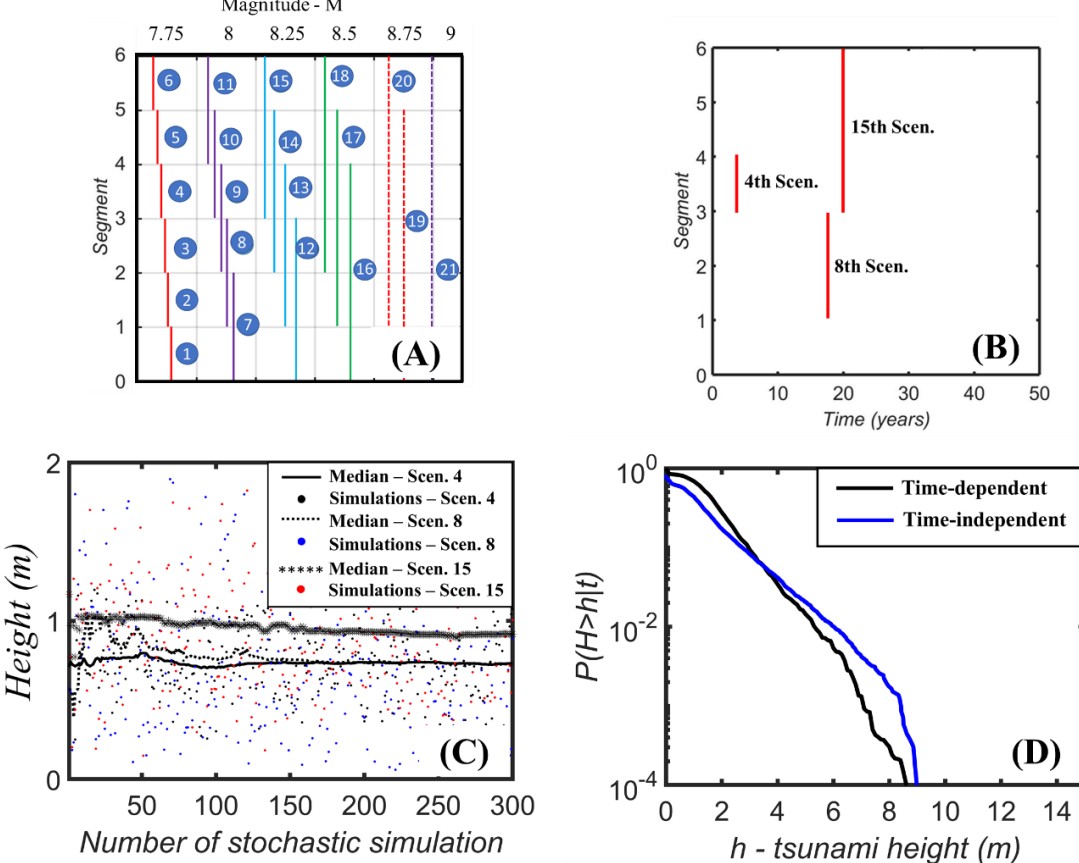

**Figure 14. Development of final hazard curve for the time-dependent model: (A) segmentation of earthquake rupture and tsunami scenarios, (B) illustration of one simulated rupture catalog (number 152) over 50 years, (C) stochastic tsunami height corresponding to the occurred scenarios in the catalog, and (D) final hazard curve.**


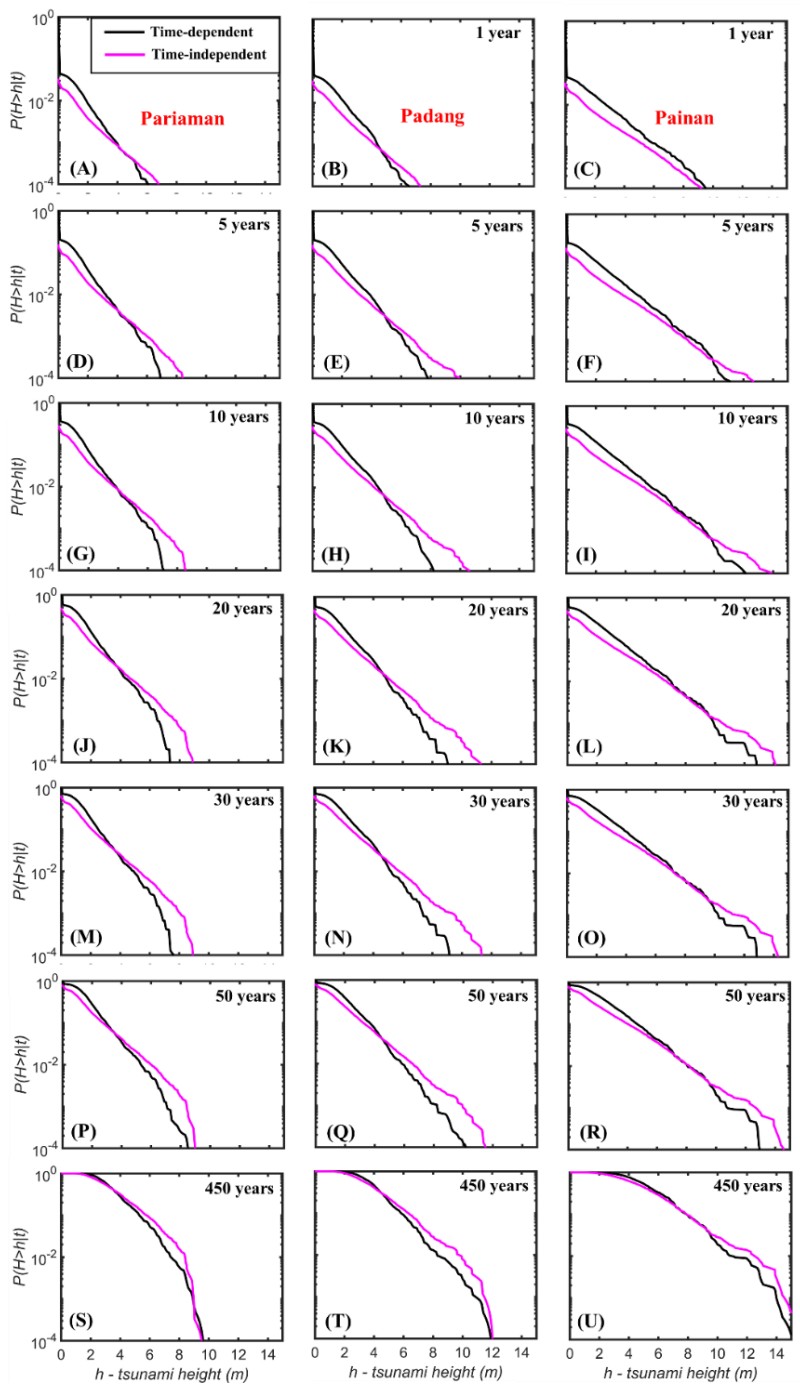

**Figure 15. Tsunami hazard curves comparison at Pariaman (left panel; P4 in Figure 8), Padang (central panel; P12 in Figure 8), and Painan (right panel; P25 in Figure 8) of time-independent PTHA and time-dependent PTHA approaches for seven considered periods: 1 year (A-C), 5 years (D-F), 10 years (G-I), 20 years (J-L), 30 years (M-O), 50 years (P-R), and 450 years (S-U).**



**Figure 16. Tsunami hazard curves comparison of 1 year along the coast of western Sumatra (P1 to P33 in Figure 8) of time-independent PTHA and time-dependent PTHA approaches.**

**Figure 17.Tsunami hazard maps near Padang (P12 in Figure 8C): (A-B) time-dependent PTHA and (C-D) time-independent PTHA (Source of base map: © Google Maps).**