# Peer review of "Time-dependent Probabilistic Tsunami Hazard Analysis for Western Sumatra, Indonesia, Using Space-Time Earthquake Rupture Modelling and Stochastic Source Scenarios"

_Natural Hazards and Earth System Sciences, 2022_

## Author Comment (AC1)

**Response to the comments of Reviewer 1**

The authors would like to thank the reviewers for their constructive and positive reviews of our manuscript. We would like to highlight several essential points that all three reviewers commonly ask:

First, our paper is a manuscript that proposes a new approach integrating the multivariate Bernoulli earthquake occurrence model and stochastic source modelling. As such, a natural scope/application of the new method is the time-dependent Probabilistic Tsunami Hazard Analysis (PTHA). We do not take a strong position as to which time-independent and time-dependent models are appropriate. However, we have noticed that the wordings in our original manuscript in supporting the time-dependent model for PTHA were relatively strong, and hence, once we are invited to revise the manuscript, we will soften our statements and clarify the motivation of the paper.

Second, the justification for the choice of the time-dependent PTHA model was not well presented in the submitted manuscript. We have prepared additional texts both in the introduction and the results and discussion sections (see details comments below) to highlight some more recent studies (i.e. Williams et al., 2019; Griffin et al., 2020; Moernaut 2020). Those studies suggest that the global paleoearthquake records provided empirical support for weakly quasiperiodic earthquake recurrence. Therefore, it can be used to justify the use of renewal models (i.e. Brownian Passage Time (BPT) model) for seismic/tsunami hazard assessment.

Third, this manuscript presents a new alternative to carrying out a time-dependent PTHA as a kind of sensitivity analysis using Indonesia (Sunda) applications. As such, we should recognize the limitations and challenges to be resolved. In our submitted manuscript, we did not provide the limitations of our study highlighted by the three reviewers. These include the 1D approach in earthquake rupture modelling, BPT parameter estimation using the Bayesian approach, and space-time earthquake rupture modelling. Subsequently, we will dedicate a new section explaining our limitations in the revised manuscript.

Fourth, a combination of multiple earthquake occurrence models can be included as a logic-tree approach. However, we did not explore this in this study as it is out of the scope. We stay on our objective: presenting a new method that combines multivariate Bernoulli and stochastic source modelling methods, which we hope to promote many applications in future.

Fifth, the valuable suggestion from the reviewers to re-do the analysis, specifically in the earthquake rupture modelling and Bayesian parameter estimation, is essential. We will re-do the earthquake rupture modelling and Bayesian parameter estimation and include the updated results in the revised manuscript:

For the earthquake rupture modelling:

- The instrumental events (i.e. the 2007a, 2007b and the 2010 events) will be excluded in modelling the time-dependent earthquake rupture.
- The earthquake catalogue data to develop a magnitude-frequency model for the time-independent approach will include the paleogeodetic records provided in Philibosian et al. (2017) by excluding the instrumental events (i.e. the 2007a, 2007b and the 2010 events). Therefore, both the time-dependent and time-independent approaches use similar earthquake data. An updated frequency-magnitude model will then be developed for the time-independent approach by fitting the Gutenberg-Richter (GR) relationship using Weichert (1980) method to treat the varying completeness magnitude. Moreover, the number of low-magnitude scenarios (i.e. < M 7.625) will be reduced by considering only the magnitude of ≥ M 6.0 (in the submitted manuscript, we use the magnitude of ≥ M 5.0).

For BPT parameters estimation using the Bayesian approach:

- The prior setup needs to be updated and adopt the uninformative priors, e.g. as used by Fitzenz et al. (2010).
- The BPT parameters estimation using the Bayesian approach will be further improved to use both the maximum a posteriori (MAP) and full posterior parametric uncertainty in defining the final BPT parameters to consider the uncertainty in BPT parameter estimation.

Furthermore, to respond to the reviewer's comments, we have copied the reviewer's comments and in italicized text.

**General Comments**

**(1) The study is based on the idea that a BPT or other time dependent rupture model more accurately represents earthquake behavior along the Sunda subduction zone. Given numerous papers refuting the seismic gap hypothesis for subduction zones in general (e.g., Rong et al., 2002 who cite Matthews, 2002), it seems that a logical first step for any study region is to falsify a Poisson null hypothesis.**

> *Thank you very much for these valuable comments. We are aware that we did not include a clear explanation regarding the selection of the time-dependent approach (i.e. BPT model) in the earthquake rupture modelling. Subsequently, we will add the following texts (in the introduction) once we are invited to revise the manuscript:*

> *In general, the earthquake rupture can be modelled using two approaches: Poisson and non-Poisson. The Poisson approach employs a memory-less Poisson process for long-term hazard assessment and is commonly adopted for earthquake rupture modelling (e.g. Burroughs and Tebbens, 2005; Tinti et al., 2005; Orfanogiannaki and Papadopoulos, 2007). However, assuming a lack*

*of memory between major earthquake occurrences is often viewed as a first approximation, inconsistent with the physics of elastic rebound (Reid, 1911; Anagnos and Kiremidjian, 1984; Berryman et al., 2012). As a result, many studies adopted a renewal model of earthquake occurrence, i.e. non-Poisson model (Matthews et al., 2002; Zhuang et al., 2012; Field et al., 2014; Williams et al., 2019; Griffin et al., 2020) to carry out a time-dependent earthquake rupture modelling. More recent studies using global paleoearthquake records (i.e. Williams et al., 2019; Griffin et al., 2020; Moernaut 2020) showed that large earthquakes in the subduction zones recur more regularly than expected from exponentially distributed interevent times (i.e., a Poisson process). Specifically, the earthquake recurrence in the Mentawai Sunda subduction zone is categorized as more like a supercycle type (i.e. a combination of large gaps and clusters), demonstrating that successive large earthquakes are dependent on each other (Salditch et al. 2020). Therefore, this study adopts the renewal model (i.e. BPT distribution) for earthquake rupture modelling in the Mentawai segment of the Sunda subduction zone.*

**(2) Although the definition of fault segments is based on 450 years of earthquake occurrence, there still might not be sufficient to determine if these segment boundaries are persistent (cf., Jackson et al, 2011).**

*Thank you very much for this comment.*

*We are aware that the current fault segment in our study may not have persistent boundaries and can be further extended. For instance, the segment can be extended to about 900 km, starting from North of Batu Island to South of Enggano Island (Natawidjaja et al., 2006; Muhari et al., 2011; Muhammad et al., 2016). Our previous studies have implemented such a model (Muhammad et al., 2016, 2017, 2018). However, in this study, to integrate the Spatio-temporal earthquake rupture modelling and stochastic tsunami simulation, we need to consider the past rupture areas to model future ruptures. Such rupture areas are developed based on the coral microatoll samples from 21 sites along 600 km of the Mentawai Islands (Natawidjaja et al., 2006; Shieh et al., 2008; Philibosian et al., 2014, 2017). Southern parts of Mentawai are not included in our model, although they could be a part of rupture areas because these areas lack of coral samples. However, this 600-km segment with a total area of 150,000 km², i.e. 600 km (length) × 250 km (width) is still consistent with the scaling relationships (e.g. Goda et al. (2016) and Thingbaijam et al. (2017)). Figure R1-1 presents the relationship between the fault areas and magnitude based on the Goda et al. (2016) and Thingbaijam et al. (2017) scaling relationships. The figure shows that the maximum fault areas used in this study represent the magnitude of between M 9.0-9.2. Therefore, it can still be adopted to constrain the boundaries of finite fault models developed in this work.*

[Figure]

*Figure R1-1. Fault areas and moment magnitude (M) relationships based on the Goda et al. (2016) and Thingbaijam et al. (2017) relationships.*

**(3) The earthquake occurrence model is based on a 1D (along strike) representation of the subduction zone. For the Sunda subduction zone, as with other subduction zones with a broad shelf, however, tsunami generation is critically dependent on the dip extent of rupture as was notably observed in comparisons of the 2004 and 2005 earthquakes (e.g., Geist et al., 2006). The limitation of the 1D approach should be mentioned.**

*Thank you very much for your valuable comment.*

*The 2D segmentation can also be possibly represented by our current setup. The first direction is along strike segmentation whereas the asperity region can be regarded as a along width segmentation as large concentration of slips is allowed in this area. This was initially considered to accommodate notable observations from the recent tsunami events, including 2011 Tohoku – large concentration of slips along the trench line. In reality, this may capture various geological environments – such as outer wedge rupture. However, such an explanation has not been included in the submitted manuscript and we will include this in the revised manuscript once we are invited to revise it.*

*On the other hand, In our setup, the strike and dip angles of the fault-plane of tsunamigenic source models are typically 296º to 326º and 7º to 19º, respectively.  These values are comparable to the slab models for the Sunda subduction zone produced by the USGS (Hayes et al., 2009, 2012). The top edge of the fault plane is located at a depth of 3 km. This depth is consistent with the past Mentawai finite-fault models developed for the 2010 Mentawai tsunamigenic earthquakes and the twin events of the 1797 and 1833, which have the top edge depth between 2 km and 5 km (Newman et al., 2011; Satake et al., 2013; Philibosian et al., 2014; Yue et al., 2014).*

*We have not explained this clearly in the manuscript, and we will include it in the manuscript once we are invited to revise the manuscript.*

**(4) It seems that it would be straightforward to estimate uncertainties in mu, alpha, and gamma from the posterior distributions (confidence intervals). These uncertainties could then be used as part of the probabilistic calculations.**

*Thank you very much for this constructive comment.*

*We understand that the BPT parameters estimation using the Bayesian approach will be further improved to consider both the maximum a posteriori (MAP) and full posterior parametric uncertainty. Therefore, we wll include this approach in the following revised manuscripts once we are invited to revise the manuscript.*

**(5) My impression is that the maximum magnitude earthquake considered is from the 450-year record and essentially is an event that spans segments 1-6. Even though the tsunami from an Mmax event would have a low probability, such an event may pose a more significant component of the aggregate hazard for longer exposure times than considered in this study. It should be clarified how Mmax is determined and whether a penultimate event could extend beyond the study region.**

*Thank you very much for this valuable comments.*

*The maximum magnitude scenario was selected based on geodetic, paleo-geodetic, and paleo-tsunami studies (Zachariasen et al., 1999; Natawidjaja et al., 2006; Sieh et al., 2008). Those studies indicated that the accumulated slip in the Mentawai segment of the Sunda subduction zone might generate tsunamigenic earthquakes ranging from M 8.8 to M 9.1. Specifically, coral microatoll samples from 21 sites along 600 km of the Mentawai region were used to constrain the dates, spatial extents and approximate earthquake source models for tsunami generation (Natawidjaja et al., 2006; Shieh et al., 2008; Philibosian et al., 2014, 2017). In our study, the tsunami simulation can not be extended into more than six segments (600 km) due to the availability of the coral microatoll samples beyond these six segments. However, this 600-km segment with a total area of 150,000 km$^2$, i.e. 600 km (length) $\times$ 250 km (width) is still consistent with the scaling relationships (e.g. Goda et al. (2016) and Thingbaijam et al. (2017)). Figure R1-2 presents the relationship between the fault areas and magnitude based on the Goda et al. (2016) and Thingbaijam et al. (2017) scaling relationships. The figure shows that the maximum fault areas used in this study represent the maximum magnitude of between M 9.0-9.2. Moreover, we allow the model to have a maximum magnitude up to about M 9.1 (+/- M 0.1 of M 9.0) in developing the stochastic earthquake source model, and hence, such a number is still sufficient to represent the maximum*

*magnitude event that may occur in the Mentawai segment of the Sunda subduction zone. Therefore, it can still be adopted to constrain the boundaries of finite fault models developed in this work.*

[Figure]

*Figure R1-2. Fault areas and M relationships based on the Goda et al. (2016) and Thingbaijam et al. (2017) relationships.*

*More importantly, we are aware that this is one of the limitations of our current work. Therefore, we will emphasize this issue in the following revised manuscript once we are invited to revise it.*

**(6) Tsunami heights seem to "saturate" at nearly 10 m (Figure 13). Is this dependent on the largest magnitude earthquake or is this caused by a hydrodynamic effect?**

*It is mainly due to the hydrodynamic effect. At other coastal points, the tsunami height may exceed 10 m. To confirm this, we plot tsunami heights along the coast of western Sumatra from all 21 scenarios, as shown in Figure R1-3. The figure shows that the tsunami height can be more than 10 m.*

[Figure]

*Figure R1-3. Tsunami height along the coast of Western Sumatra from the 21 scenarios adopted in this study.*

**In-line comments**

*To respond to the reviewer's in-line comments, we have copied the reviewer's comments and then replied in italicized text.*

L42: Vere-Jones' stress release model (cf., Bebbington and Harte, 2001) could also be mentioned—more relevant to this study. Moreover, Eqn. 3 is a cumulative distribution function, not a frequency-magnitude distribution.

>*We will make the change based on your comments in the revised manuscript.*

L141: I couldn't find in the manuscript where the specific magnitude-area relation used was mentioned. Since this is often a contentious choice, especially for subduction zone earthquakes, the specific relation and its justification should be indicated.

>*In the stochastic source model generation, we only consider the length and width generated based on the scaling relationships developed by Goda et al. (2016). However, we used magnitude-area relation to constrain the generated parameters in the final checking of the generated parameters. We have confirmed that the generated area of the finite-fault model is consistent with the adopted scaling relationships.*

>*We will add this explanation once we are invited to revise the manuscript.*

L257: How is distance D determined?

>*A length of 100 km corresponds to a minimum magnitude (M 7.75) of a significant tsunami-earthquake adopted in this study. Such a value is consistent with the length of fault developed by Goda et al. (2016) that is used to generate the stochastic earthquake source modelling.*

L316: Same variable D used for slip here and distance in L257.

Fig. 3: "occurred scenarios" is awkward. Could just say "scenarios".

>*We will change these terms in L316 and Fig. 3 in the revised manuscript.*

**References**

Fitzenz, D. D., Ferry, M. A., & Jalobeanu, A. (2010). Long-term slip history discriminates among occurrence models for seismic hazard assessment. Geophysical Research Letters, 37(20), 1–5. https://doi.org/10.1029/2010GL044071.

Goda, K., Yasuda, T., Mori, N., and Maruyama, T. (2016). New scaling relationships of earthquake source parameters for stochastic tsunami simulation. Coastal Eng. J.. doi:10.1142/S0578563416500108.

Griffin, J. D.; Stirling, M. W. & Wang, T. Periodicity and Clustering in the Long-Term Earthquake Record Geophysical Research Letters, American Geophysical Union (AGU), 2020, 47.

Hayes, G.P., Wald, D.J., and Keranen, K. (2009). Advancing techniques to constrain the geometry of the seismic rupture plane on subduction interfaces a priori - higher order functional fits, *Geochem. Geophys. Geosyst*. 10, Q09006, doi:10.1029/2009GC002633.

Hayes, G.P., Wald, D.J., and Johnson, R. L. (2012), Slab1.0: A three-dimensional model of global subduction zone geometries, J. Geophys. Res. 117, B01302, doi:10.1029/2011JB008524.

Moernaut, J. (2020, November 1). Time-dependent recurrence of strong earthquake shaking near plate boundaries: A lake sediment perspective. Earth-Science Reviews. Elsevier B.V. https://doi.org/10.1016/j.earscirev.2020.103344.

Natawidjaja, D.H., Sieh, K., Chlieh, M., Galetzka, J., Suwargadi, B.W., Cheng, H., Edwards, R.L., Avouac, J.P., and Ward, S.N. (2006). Source parameters of the great Sumatran megathrust earthquakes of 1797 and 1833 inferred from coral microatolls. J. Geophys. Res. 111, B06403, doi:10.1029/2005JB004025.

Newman, A.V., Hayes, G., Wei, Y., and Convers, J. (2011). The 25 October 2010 Mentawai tsunami earthquake, from real-time discriminants, finite-fault rupture, and tsunami excitation. Geophys. Res. Lett. 38, 1–7. doi:10.1029/2010GL046498.

Muhammad, A., Goda, K., & Alexander, N. (2016). Tsunami hazard analysis of future megathrust sumatra earthquakes in Padang, Indonesia using stochastic tsunami simulation. Frontiers in Built Environment, 2, 33.

Muhammad, A., Goda, K., Alexander, N. A., Kongko, W., & Muhari, A. (2017). Tsunami evacuation plans for future megathrust earthquakes in Padang, Indonesia, considering stochastic earthquake scenarios. Natural Hazards and Earth System Sciences, 17(12), 2245-2270.

Muhammad, A., & Goda, K. (2018). Impact of earthquake source complexity and land elevation data resolution on tsunami hazard assessment and fatality estimation. Computers & geosciences, 112, 83-100.

Philibosian, B., Sieh, K., Avouac, J.P, Natawidjaja, D.H., Chiang, H., Wu, C., Perfettini, H., Shen, C.C., Daryono, M.R., and Suwargadi, B.W. (2014). Rupture and variable coupling behavior of the Mentawai segment of the Sunda megathrust during the super cycle

culmination of 1797 to 1833. J. Geophys. Res. Solid Earth. 119, 7258–7287, doi:10.1002/2014JB011200.

Philibosian, B., Sieh, K., Avouac, J. P., Natawidjaja, D. H., Chiang, H. W., Wu, C. C., ... & Wang, X. (2017). Earthquake supercycles on the Mentawai segment of the Sunda megathrust in the seventeenth century and earlier. Journal of Geophysical Research: Solid Earth, 122(1), 642-676.

Satake, K., Nishimura, Y., Putra, P. S., Gusman, A.R., Sunendar, H., Fujii, Y., Sunendar, H., Latief, H., and Yulianto, E. (2013). Tsunami source of the 2010 Mentawai, Indonesia earthquake inferred from tsunami field survey and waveform modeling. Pure Appl. Geophys. 170, 1567–1582. doi:10.1007/s00024-012-0536-y.

Sieh, K., Natawidjaja, D.H., Meltzner, A.J., Shen, C.C., Cheng, H., Li, K.S., Suwargadi, B.W., Galetzka, J., Philibosian, B., and Edwards, R.L. (2008). Earthquake super cycles inferred from sea-level changes recorded in the corals of West Sumatra. Science 322, 1674–1678. doi:10.1126/science.1163589.

Thingbaijam, K. K. S., Mai, P. M., & Goda, K. (2017). New Empirical Earthquake Source-Scaling LawsNew Empirical Earthquake Source-Scaling Laws. Bulletin of the Seismological Society of America, 107(5), 2225-2246.

Weichert, D. H. (1980). Estimation of the earthquake recurrence parameters for unequal observation periods for different magnitudes. Bulletin of the Seismological Society of America, 70(4), 1337-1346.

Williams, R. T., Davis, J. R., & Goodwin, L. B. (2019). Do Large Earthquakes Occur at Regular Intervals Through Time? A Perspective From the Geologic Record. Geophysical Research Letters, 46(14), 8074–8081. https://doi.org/10.1029/2019GL083291.

Yue, H., Lay, T., Rivera, L., Bai, Y., Yamazaki, Y., Cheung, K.F., Hill, E.M., Sieh, K., Kongko, W., and Muhari, A. (2014). Rupture process of the 2010 Mw7.8 Mentawai tsunami earthquake from joint inversion of near-field hr-GPS and teleseismic body wave recordings constrained by tsunami observations. J. Geophys. Res. Solid Earth. 119, 5574–5593. doi:10.1002/2014JB011082.

Zachariasen, J., Sieh, K., Taylor, F.W., Edwards, R.L., and Hantoro, W.S. (1999). Submergence and uplift associated with the giant 1833 Sumatran subduction earthquake: Evidence from coral microatolls. J. Geophys. Res. 104, 895–919.

---

## Author Comment (AC2)

**Response to the comments of Reviewer 2**

The authors would like to thank the reviewers for their constructive and positive reviews of our manuscript. We would like to highlight several essential points that all three reviewers commonly ask:

First, our paper is a manuscript that proposes a new approach integrating the multivariate Bernoulli earthquake occurrence model and stochastic source modelling. As such, a natural scope/application of the new method is the time-dependent Probabilistic Tsunami Hazard Analysis (PTHA). We do not take a strong position as to which time-independent and time-dependent models are appropriate. However, we have noticed that the wordings in our original manuscript in supporting the time-dependent model for PTHA were relatively strong, and hence, once we are invited to revise the manuscript, we will soften our statements and clarify the motivation of the paper.

Second, the justification for the choice of the time-dependent PTHA model was not well presented in the submitted manuscript. We have prepared additional texts both in the introduction and the results and discussion sections (see details comments below) to highlight some more recent studies (i.e. Williams et al., 2019; Griffin et al., 2020; Moernaut 2020). Those studies suggest that the global paleoearthquake records provided empirical support for weakly quasiperiodic earthquake recurrence. Therefore, it can be used to justify the use of renewal models (i.e. Brownian Passage Time (BPT) model) for seismic/tsunami hazard assessment.

Third, this manuscript presents a new alternative to carrying out a time-dependent PTHA as a kind of sensitivity analysis using Indonesia (Sunda) applications. As such, we should recognize the limitations and challenges to be resolved. In our submitted manuscript, we did not provide the limitations of our study highlighted by the three reviewers. These include the 1D approach in earthquake rupture modelling, BPT parameter estimation using the Bayesian approach, and space-time earthquake rupture modelling. Subsequently, we will dedicate a new section explaining our limitations in the revised manuscript.

Fourth, a combination of multiple earthquake occurrence models can be included as a logic-tree approach. However, we did not explore this in this study as it is out of the scope. We stay on our objective: presenting a new method that combines multivariate Bernoulli and stochastic source modelling methods, which we hope to promote many applications in future.

Fifth, the valuable suggestion from the reviewers to re-do the analysis, specifically in the earthquake rupture modelling and Bayesian parameter estimation, is essential. We will re-do the earthquake rupture modelling and Bayesian parameter estimation and include the updated results in the revised manuscript:

For the earthquake rupture modelling:

- The instrumental events (i.e. the 2007a, 2007b and the 2010 events) will be excluded in modelling the time-dependent earthquake rupture.
- The earthquake catalogue data to develop a magnitude-frequency model for the time-independent approach will include the paleogeodetic records provided in Philibosian et al. (2017) by excluding the instrumental events (i.e. the 2007a, 2007b and the 2010 events). Therefore, both the time-dependent and time-independent approaches use similar earthquake data. An updated frequency-magnitude model will then be developed for the time-independent approach by fitting the Gutenberg-Richter (GR) relationship using Weichert (1980) method to treat the varying completeness magnitude. Moreover, the number of low-magnitude scenarios (i.e. < M 7.625) will be reduced by considering only the magnitude of ≥ M 6.0 (in the submitted manuscript, we use the magnitude of ≥ M 5.0).

For BPT parameters estimation using the Bayesian approach:

- The prior setup needs to be updated and adopt the uninformative priors, e.g. as used by Fitzenz et al. (2010).
- The BPT parameters estimation using the Bayesian approach will be further improved to use both the maximum a posteriori (MAP) and full posterior parametric uncertainty in defining the final BPT parameters to consider the uncertainty in BPT parameter estimation.

Furthermore, to respond to the reviewer's comments, we have copied the reviewer's comments and in italicized text.

**High-level Comments**

**P1: Generally the paper argues that time-dependent modelling is more accurate, but doesn't provide strong justification for this. To my knowledge there are contrasting views on this in the literature, which should be represented in this paper. The language should be softened, and uncertainties better discussed (see detailed comments).**

*We thank you for these valuable comments. Subsequently, we will add the following texts (in the Introduction) once we are invited to revise the manuscript:*

*In general, the earthquake rupture can be modelled using two approaches: Poisson and non-Poisson. The Poisson approach employs a memory-less Poisson process for long-term hazard assessment and is commonly adopted for earthquake rupture modelling (e.g. Burroughs and Tebbens, 2005; Tinti et al., 2005; Orfanogiannaki and Papadopoulos, 2007). However, assuming a lack of memory between major earthquake occurrences is often viewed as a first approximation, inconsistent with the physics of elastic rebound (Reid, 1911; Anagnos and Kiremidjian, 1984; Berryman et al., 2012). As a result, many studies adopted a renewal*

*model of earthquake occurrence, i.e. non-Poisson model (Matthews et al., 2002; Zhuang et al., 2012; Field et al., 2014; Williams et al., 2019; Griffin et al., 2020) to carry out a time-dependent earthquake rupture modelling. More recent studies using global paleoearthquake records (i.e. Williams et al., 2019; Griffin et al., 2020; Moernaut 2020) showed that large earthquakes in the subduction zones recur more regularly than expected from exponentially distributed interevent times (i.e., a Poisson process). Specifically, the earthquake recurrence in the Mentawai Sunda subduction zone is categorized as more like a supercycle type (i.e. a combination of large gaps and clusters), demonstrating that successive large earthquakes are dependent on each other (Salditch et al. 2020). Therefore, this study adopts the renewal model (i.e. BPT distribution) for earthquake rupture modelling in the Mentawai segment of the Sunda subduction zone.*

*Moreover, across the content of the revised manuscript, we will soften the term of 'the time-dependent modelling is more accurate' to consider the reviewers' comments.*

**P2: The maximum-magnitude is set to Mw9. In reality maximum magnitudes are quite uncertain (justified further below) and yet very impactful for the results. Again I expect they are likely more important than the effect of time-dependence in the current modelling. Many other PTHA studies treat this as an uncertain parameter (details below), and I suggest that issue is also addressed in this paper.**

*Thank you very much for your comments.*

*In this study, to integrate the spatio-temporal time-dependent earthquake rupture modelling, we adopt a 1D along-strike distance discretized into six segments. Each segment represents the smallest area that may rupture in tsunamigenic earthquakes. Such a 1D along-strike model is developed based on the past fault rupture, and the total distance of these 6 segments (i.e. 600 km) is used to represent the maximum magnitude scenario.*

*The maximum magnitude scenario was selected based on geodetic, paleo-geodetic, and paleo-tsunami studies (Zachariasen et al., 1999; Natawidjaja et al., 2006; Sieh et al., 2008). Those studies indicated that the accumulated slip in the Mentawai segment of the Sunda subduction zone might generate tsunamigenic earthquakes ranging from M 8.8 to M 9.1. Specifically, coral microatoll samples from 21 sites along 600 km of the Mentawai region were used to constrain the dates, spatial extents and approximate earthquake source models for tsunami generation (Natawidjaja et al., 2006; Sieh et al., 2008; Philibosian et al., 2014, 2017). In our study, the tsunami simulation can not be extended into more than six segments (600 km) due to the availability of the*

*coral microatoll samples beyond these six segments. However, this 600-km segment with a total area of 150,000 km², i.e. 600 km (length) × 250 km (width) is still consistent with the scaling relationships (e.g. Goda et al. (2016) and Thingbaijam et al. (2017)). Figure R2-1 presents the relationship between the fault areas and magnitude based on the Goda et al. (2016) and Thingbaijam et al. (2017) scaling relationships. The figure shows that the maximum fault areas used in this study represent the maximum magnitude of between M 9.0-9.2. Moreover, we allow the model to have a maximum magnitude up to about M 9.1 (+/- M 0.1 of M 9.0) in developing the stochastic earthquake source model, and hence, such a number is still sufficient to represent the maximum magnitude event that may occur in the Mentawai segment of the Sunda subduction zone. Therefore, it can still be adopted to constrain the boundaries of finite fault models developed in this work.*

[Figure]

*Figure R2-1. Fault areas and M relationships based on the Goda et al. (2016) and Thingbaijam et al. (2017) relationships.*

*More importantly, we are aware that this is one of the limitations of our current work. Therefore, we will emphasize this issue in the following revised manuscript once we are invited to revise it.*

**P3: In so far as I can tell, there are a number of technical weaknesses in the scenario frequency modelling that should be addressed or clarified.**

**The tsunami hazard results do not seem account for uncertainties in the scenario frequency model parameters (e.g. b, rate of earthquakes, maximum-magnitude, and other parameters). This is true for both the time-dependent and time-independent models, although details of their parameters are different. Variation of the model parameters within the statistical uncertainties will likely have a substantial impact on the results, especially given only 10 events have been used to constrain the time-dependent model (which has many parameters). When these uncertainties are accounted for, I expect they will be larger than the current difference between the time-dependent and time-independent results. Given the 'many synthetic catalogues' approach used in**

**this paper, the uncertainties could be accounted for by randomly drawing different model parameters for each synthetic catalogue.**

**+ The long-term data (10 events over the last 450 years) likely has a time-varying magnitude of completeness; the earliest 8 events all have Mw>=8.3, while only the most recent 2 events (2007+) have Mw<=8. This is not surprising - a-priori we certainly expect that it would be harder to detect smaller earthquakes in the paleo data. But the statistical methods seem to ignore this issue. This could have a large impact on the fit of the time-dependent model.**

**+ The time-independent model is fit to different data than the time-dependent model, and the time-independent fit is dominated by small earthquakes (mostly having magnitudes well below the Mw 7.65+ that are of interest in this study). Even if there were no differences between the models, the use of such different data would lead to differences in their results. This makes it hard to determine the significance of the time-dependent model structure for the PTHA results. To remedy this, the long-term data should be used to constrain the time-independent model for larger magnitudes. This may require accounting for time-variations in the catalogue completeness (citations below), and placing less weight on low magnitude earthquakes (so they don't dominate the fit at higher magnitudes, which currently doesn't agree especially well with the data). This should help reduce the under-estimation of the earthquake frequencies with Mw>=8.3 (currently about three times less common in the time-independent model, vs the long term data).**

**+ The fit of the time-dependent model seems to ignore the change in completeness magnitude of the long-term data.**

> *Thank you very much for these valuable suggestions.*
>
> *We realized that we have a few technical weaknesses in developing the earthquake rupture modelling, including:*
>
> *(1) Use different catalogue data to model the magnitude frequency relationship between the time-dependent and time-independent models. The time-independent model adopts only the earthquake catalogue data from 1970, whilst the time-dependent data consider solely the paleogeodetic records from the sixteenth century.*
> *(2) Consider a large number of small earthquakes (i.e. M 5.5 – M 6.0) for the time-independent model that is well below the minimum magnitude considered in this study (i.e. M 7.625).*
> *(3) Include the instrumental events (i.e. the 2007a, 2007b and the 2010 events) for the time-dependent model that the paleogeodetic records can not detect.*
>
> *Subsequently, the following revisions will be conducted in the revised manuscript once we are invited:*

*First, the instrumental events (i.e. the 2007a, 2007b and the 2010 events) will be excluded in modelling the time-dependent earthquake rupture.*

*Second, the earthquake catalogue data to develop a magnitude-frequency model for the time-independent approach will include the paleogeodetic records provided in Philibosian et al. (2017) by excluding the instrumental events (i.e. the 2007a, 2007b and the 2010 events). Therefore, both the time-dependent and time-independent approaches use similar earthquake data. An update frequency-magnitude model will then be developed for the time-independent approach by fitting the Gutenberg-Richter (GR) relationship using Weichert (1980) method to treat the varying completeness magnitude. Moreover, the number of low-magnitude scenarios (i.e. < M 7.625) will be reduced by considering only the magnitude of ≥ M 6.0 (in the submitted manuscript, we use the magnitude of ≥ M 5.0).*

*Consequently, we perform the initial modelling of the magnitude frequency relationship by considering the new earthquake catalogue (i.e. integration catalogue) to understand the difference in the magnitude frequency distribution before and after the integration of paleogeodetic records. The results show that the b value produced from the Weichert (1980) approach is far less than the previous model, i.e. $b = 0.72$ for the Weichert model vs $b = 1.05$ for the catalogue excluding the paleogeodetic record. Furthermore, we plot the probability distribution of magnitude from the non-integrated (excluding the paleogeodetic records) and the integrated catalogues (including the paleogeodetic records) as shown in Figure R2-2. The figure clearly shows a significant change in magnitude probability for low (< M 8.0) and high (> M 8.3) scenarios. The probability of low magnitude in the integrated catalogue is almost twice smaller than the non-integrated catalogue. In contrast, the probability of the high magnitude of the integrated catalogue is twice higher than that of the-non integrated catalogue. Such a change will definitely influence the final hazard curve of the time-independent model.*

[Figure]

[Figure]

*Figure R2-2. Probability magnitude distribution produced from the non-integrated catalogue (a) and the integrated catalogue (b).*

*The modified results for comparing time-dependent and time-independent PTHA will be included in the revised manuscript, specifically in the methodology, results and discussions sections. We will run the new earthquake rupture modelling for the time-dependent case by excluding the instrumental events and update the results accordingly once we are invited to revise the manuscript.*

**+ There appear to be some anomalies in the Bayesian fit of the time-dependent model. The priors for some parameters seem to be set using the same data used for fitting, which should not be done with Bayesian statistics. Also, the figures show differences between the priors and posteriors that suggest a poor specification of the priors (details in comments below).**

*Thank you very much for these valuable comments.*

*We agree with the reviewer's comments and will re-analyze the Bayesian estimation by updating the following steps:*

- *The prior setup needs to be updated and adopt the uninformative priors, e.g. as used by Fitzenz et al. (2010).*
- *The BPT parameters estimation using the Bayesian approach will be further improved to use both the maximum a posteriori (MAP) and full posterior parametric uncertainty in defining the final BPT parameters to consider the uncertainty in BPT parameter estimation.*

**Detailed Comments**

**Near L25: 'Over the next decades, major tsunamigenic events are anticipated in .... '. It sounds like "we expect large tsunamis in each of these subduction zones within a few decades". I don't think this is well justified. Do the references really backup the 'major events in the next-few-decades' claim? Historically, time-dependent predictions over these kinds of timescales have not performed well for subduction zones (e.g. Rong et al., 2003).**

*We thank the reviewer for this comment.*

*We are aware that a statement of 'possible major tsunamigenic event in the future may occur over the next few decades' may be over-claimed. On the other hand, the recent studies by Philibosian et al. (2014, 2017) suggested that the next rupture of the Mentawai segments will occur in the next few decades. Such a finding is based on the past earthquake supercycles data taken from coral microatol samples since the 14th century. The rupture cascades on the Mentawai segment occur approximately every 200 years, with the last sequence of significant tsunamigenic events being the 1979 and 1833 events.*

*Moreover, the data show that a 40-year gap is found between the two largest events in each sequence (i.e. 1350/1388, 1658/1703, and 1797/1833). The intervals between other events during the 1500s and 1600s were all less than 40 years. With the last significant event occurring in 2007/2008, the next second event is expected to be in a sequence within the next few decades (i.e. ~ 40 years) and may occur within the next few decades (Philibosian et al., 2017).*

*However, we are aware that we do not discuss other findings (e.g. Rong et al., 2003) which counterargue the abovementioned statement. Therefore, we will soften the sentence in the revised manuscript to consider the uncertainty of such estimation as suggested by the reviewer.*

**Near L40: "...assuming a lack of memory between major earthquake occurrences is often viewed as a first approximation .." -- I think there are contrasting views on this in the literature, that should be represented in this part of the paper. For example Rong et al. (2003) are quite critical of assumed quasi-periodic earthquake recurrence (on empirical grounds). OTOH there is empirical evidence that large earthquakes tend to be weakly periodic, but without correlation between successive inter-event times (Griffin et al., 2020).**

*Thank you very much for this valuable suggestion. Subsequently, we will add the following texts (in the introduction) once we are invited to revise the manuscript:*

> *In general, the earthquake rupture can be modelled using two approaches: Poisson and non-Poisson. The Poisson approach employs a memory-less Poisson process for long-term hazard assessment and is commonly adopted for earthquake rupture modelling (e.g. Burroughs and Tebbens, 2005; Tinti et al., 2005; Orfanogiannaki and Papadopoulos, 2007). However, assuming a lack of memory between major earthquake occurrences is often viewed as a first approximation, inconsistent with the physics of elastic rebound (Reid, 1911; Anagnos and Kiremidjian, 1984; Berryman et al., 2012). As a result, many studies adopted a renewal model of earthquake occurrence, i.e. non-Poisson model (Matthews et al., 2002; Zhuang et al., 2012; Field et al., 2014; Williams et al., 2019; Griffin et al., 2020) to carry out a time-dependent earthquake rupture modelling. More recent studies using global paleoearthquake records (i.e. Williams et al., 2019; Griffin et al., 2020; Moernaut 2020) showed that large earthquakes in the subduction zones recur more regularly than expected from exponentially distributed interevent times (i.e., a Poisson process). Specifically, the earthquake recurrence in the Mentawai Sunda subduction zone is categorized as more like a supercycle type (i.e. a combination of large gaps and clusters), demonstrating that successive large earthquakes are dependent on each other (Salditch et al. 2020).*

*Therefore, this study adopts the renewal model (i.e. BPT distribution) for earthquake rupture modelling in the Mentawai segment of the Sunda subduction zone.*

**L53-54: "Recent work has also used high-resolution spatial grids .... to produce more accurate tsunami hazard results (e.g. < 90m, ...)". I don't think we should describe "<90m" as high-resolution for onshore work, that is quite coarse. I might describe resolutions of 10m or less as high-resolution (e.g. Gibbons et al., 2020).**

*Thank you for this comment, and we agree that less than 10 m is a high-resolution grid. We will update the sentences when we are invited to revise the manuscript.*

**Near L67: "Since time-dependent hazard estimation leads to more realistic short-term results" -- this really needs justification, or removal. To my knowledge this point has not been demonstrated in general, and it may-or-may-not be true. I suppose for aftershock modelling there would be lots of evidence, but this study is using quasi-periodic modelling for large events, and I believe there is less evidence on this matter. In the Paleo record, some sites look more time-dependent than others, e.g. Griffin et al. 2020.**

*Thank you for this comment, and we will remove it from the manuscript once we are invited to revise it.*

**Near L74: "A uniform-slip was used, which may underestimate the hazard..." -- I believe Horspool et al. (2014) used a log-normal distribution to predict the (uncertain) heights at the coast from the uniform-slip scenarios, as a way of accounting for uncertainties due to the slip model and uncertain geometry. In principle this is supposed to compensate for the lack of slip heterogeneity. In practice it could either underestimate, or overestimate, the variability of natural earthquake-tsunamis. If their sigma were sufficiently large, it may even predict greater hazard than your model (I haven't checked whether it does, just clarifying the principle).**

*Thank you very much for this critical comment.*

*We wrote such a statement because our previous study (i.e. Muhammad et al. 2018) has compared the tsunami hazard and risk assessment results between the uniform and stochastic source models. The study found that the uniform model significantly underestimates the tsunami hazards compared to the stochastic source models.*

*However, we will update the wording within the mentioned statement to consider the possibility of under-estimation/overestimation of the hazards from the work of Horspool et al. (2014).*

**L86: Please add a statement about why you use segments to define the rupture extents (I think it is related to the space-time modelling?).**

*The segments are used to integrate the spatio-temporal time-dependent earthquake rupture modelling. We adopt a 1D along-strike distance discretized into six segments with the length of each segment of 100 km. One discretized segment represents the smallest areas that may rupture in tsunamigenic earthquakes. Moreover, the total distance of these six segments is used to represent the maximum magnitude scenario.*

**L115: "magnitude-frequency distribution" -- Should this be "probability density function"? I think the MFD would include the factor lambda_i.**

*We agree with the reviewer's comment. This is a continuous probability density function. In our study, we approximated the continuous $f(M)$ probability function with a discrete magnitude function $P(M_j)$ $(j = 1, ..., n)$ of $n$ bins with bin width $\Delta M$ and hence, we mention this term as the magnitude-frequency distribution.*

**L117: "frequency-magnitude distribution" -- It think this should be "Cumulative Distribution Function"? Furthermore I think you need to say that f_i(M) is the derivative of F(M) (and consider whether you need a subscript _i for F).**

**L125: In Equation 4, the subscript '_i' might be confused with the same subscript used to denote the source in Equation 2. Also, I think Eq 4 should use 'j' for consistency with notation in the paragraph just before Equation 4?**

*Thank you very much for your suggestion. We will change the terms for both lines in the revised manuscript if we are invited to revise it.*

**L127: Here I am concerned that you are not using the long-term paleo data to fit the GR model. Why not? The longer term data suggests a high rate of Mw >= 8.3 (8 events in 450 years, rate around 0.018), quite a bit more frequent than suggested by your time-independent model (visually seems ~ 0.006 in Figure 1C, or one-third the frequency -- noting this fit is dominated by low-magnitude earthquakes, below magnitudes of practical interest for this study). As well as taking the opportunity to improve the model accuracy, this would be good because the long-term data is used for fitting the time-dependent model. The use of very different data to fit the two models allows for a substantial 'arbitrary' difference between their results, which is not related to their structure (temporal/non-temporal). I am concerned that this may dominate the differences in your results. I would suggest you fit the time-independent GR model using both the long-term and catalogue data (there are various approaches to treating the varying completeness magnitude, e.g. Weichert, 1980), while removing the instrumental events from the long-term data. Also, you might want to use fewer low-magnitude earthquakes to constrain the fit (to**

**reduce the influence of low-magnitude earthquakes on the fit, and better represent the data at magnitudes that matter for this PTHA).**

*Thank you very much for your comments. This issue is similar to your opinion in 'High-Level Comments'. We have discussed this issue in point number 3 above.*

**L135: Around here, could you please explicitly state that the time-dependent model does not have Mw-frequency curves that follow the GR distribution, over any time-scale. I didn't realize this initially, and it is obviously a very important point for the subsequent analysis. Perhaps a sentence highlighting that instead the Mw-frequency distribution will reflect correlations between rupture on different segments, which is parameterized by the model itself.**

*We thank you for this suggestion and agree that our time-dependent model does not use the same Mw-frequency distribution following the GR distribution as we model the time-independent approach. We will add additional sentences to clarify this issue in the revised manuscript.*

**L150: It looks like the magnitudes only go up to 9? I think this is neglecting the large uncertainties in Mw-max. Neglect of those uncertainties may have a strong impact on the results. A few relevant points: Berryman et al. (2015) suggested uncertain Mw-max values in this region ranging from 9.0 - 9.6 based on scaling relations and the historical record. Such highly uncertain Mw-max values have been represented in PTHAs (e.g. Davies et al., 2017; Davies and Griffin, 2020). Horspool et al. (2014) allowed Mw-max on Sumatra to vary in 9.3 - 9.7. We know the nearby 2004 event had a magnitude exceeding 9 (around 9.2). From Tohoku we also know that Mw 9.1 can occur in relatively compact regions, smaller than the extents of your study. On this basis I don't think we can exclude the possibility of higher magnitude earthquakes.**

*Thank you very much for your comments. We have discussed this issue in point number 3 of responding to your 'High-Level Comments'.*

**L153: "..for each of those 21 rupture scenarios" -- suggest to add "geometrical" before "rupture scenarios", to be consistent with previous sentences. Here there are a few interacting concepts: "geometrical rupture scenarios (seems to be a magnitude plus a set of segments?)", "scenarios", "events" (is this the same as "scenarios"). I suggest you pick one term for each concept, and then use it consistently throughout the paper.**

*Thank you for this valuable suggestion. This 'rupture scenarios' may also explain both the magnitude and a set of segments; hence, we will add the geometrical before the rupture scenarios. We are also aware that the used terms in our manuscripts were not consistent. We will modify the mentioned terms (e.e. scenarios and events) to be more consistent in the revised manuscript.*

**L 164: "(one height for one simulation catalog)" -- does the height vary with space, or are we looking at the 'maximum height anywhere in the model'?**

*The height mentioned here is varied in space, meaning that the tsunami height differs from one location to another. Figure R2-3 shows the tsunami heights along the western coast of Sumatra generated from the 300 stochastic models of M 7.75. The figure clearly shows that at each point, the height is varied depending on the earthquake source model parameters and the location.*

[Figure]

*Figure R2-3. Tsunami height along the west coast of Sumatra.*

*Moreover, the height varies depending on the stochastic models. Therefore, each stochastic model produces a different height level due to different earthquake source parameters, particularly the slip distribution. We had 300 different height levels for each rupture scenario taken from the 300 stochastic earthquake source models.*

**L 172: "The results confirm that N_{sim} = 100,000 catalogues are sufficient to produce a stable result" -- stable in terms of what? The mean over all catalogues? Please make this clear, as I suppose individual catalogues must vary greatly.**

*Thank you for this question.*

*We are aware that we did not explain this issue clearly. The stable result here refers to the relative frequencies of the 21 rupture scenarios. **Error! Reference source not found.**R2-4 shows the relative frequency of each of the 21 scenarios. The figure is plotted as a function of the simulated catalogues over the seven target periods, i.e. 1, 5, 10, 20, 30, 50, and 450 years, where the simulation started in 2021. The results show that $N_{sim}$ = 100,000 catalogs are sufficient to produce a stable result, and hence, we use $N_{sim} \geq 100,000$, varied for different periods.*

[Figure]

*Figure R2-4. Relative frequency of the 21 scenarios for different numbers of simulations.*

**L175-179: This section is confusing me. Above I understood that you used N_{sim} = 100,000 to get a stable result. But now it is suggested that many more catalogues were required for 1-50 years. Please edit to make this clearer. [NOTE: Some sentences from the 'Results' section may help in this regard, mentioned below.].**

*Thank you very much for this comment. We have explained briefly in the manuscript about this issue and will add the following additional texts to explain this:*

*The need of more number of simulations for shor periods (1-50 years) i.e. >100,000 catalogue is to obtain sufficient results at the probability level of $10^{-4}$. The probability level of $10^{-4}$ is in general used to represent the return period of several thousand (e.g. 2,500 years) relevant for tsunami hazard mapping purposes as one of the main goals we want to evaluate in this study. Subsequently, we adopt the 10,000,000 catalogues for 1-year earthquake rupture simulations because it is sufficient to obtain the hazard results at the probability level of $10^{-4}$. The number of simulations is then reduced by 10 at longer durations to minimize the computational cost, i.e. a total of 1,000,000 and 100,000 simulation numbers are used for 5 to 50 years and 450 years, respectively. Such numbers are enough to consider the probability level representing the return period of several thousand years (e.g. 2,500 years).*

**L197-199: "This number is consistent with the GR model". In my judgement they are "not very consistent", with the model under-predicting the frequency of large events (as discussed above, the GR model has a substantially lower frequency of Mw>=8.3). Note the 450 year record contains 10 events (Mw 7.8-8.9), but the first 8 events have Mw>=8.3, and the last two events are from the recent instrumental period. This suggests changes in the magnitude of completeness of the 450 year catalogue over time. A-prior we expect this would happen because Paleo records find it more difficult to detect small events. This issue should be accounted for when comparing the GR model with the long term data (and above I suggest that the long-term data should also be used to fit the time-independent GR model -- doing that will probably lead to significant increases in the modelled frequency of large earthquakes).**

*Thank you very much for your comments. We have discussed this issue in point number 3 (P3) of responding to your 'High-Level Comments'.*

**L205: "see Figure and Figure 5" -- missing Figure number.**

**L217: Suggest you use a word other than "scenarios" to denote the 21 "magnitude + set of-segments" combinations.**

*Thank you very much for your suggestion. We will change the terms for both lines in the revised manuscript if we are invited to revise it.*

**L250-ish: Above I argued that the long-term data (10 events, 450 years) is likely subject to a varying completeness magnitude, noting the only two events with Mw<8 events are recent instrumental events, and all others have Mw>=8.3. From what I can see, this 'changing completeness magnitude' is not accounted for in the statistical fit of the time-varying model (Section 2.2.2). I expect this would have a large effect on time-varying model fit - for example, overestimating the conditional probability of multi-segment rupture (which also effects the frequency of high Mw events), and affecting the BPT model parameters.**

*Thank you very much for your comments. We discussed this issue in point 4 of responding to your 'High-Level Comments'.*

**L250: "The prior median of mu for each segment is different, namely ....... These values represent the median interarrival time of earthquake rupture on each segment over the last 450 years". It sounds like you are using the same data both to specify the priors, and then to fit the model (?). In Bayesian statistics, the priors should be specified in a way that doesn't use the fitting data, or at least doesn't use it in important ways. Another potential problem with the methodology is suggested by Figure 9, where we see the prior and posterior for 'mu' are very different on some segments -- the posterior is more diffuse and often has a very different average (e.g. Panels A, I, K). This suggests the priors have been overly constrained in the analysis. Typically priors would be set either using data different used for fitting, or given weakly informative values.**

*We agreed with the reviewer's comment and will re-analyze the Bayesian estimation. The prior will be updated and adopt the uninformative priors (e.g. as used by Fitzenz et al 2010).*

**L310: This source zone has some history of "tsunami-earthquakes", with waves much larger than might be expected from the magnitude (e.g. Mentawai 2010). Can the current model produce similar large waves for scenarios with magnitude below 8, using the rigidity of 40GPA? I would be surprised if it can, although that will also depend on how concentrated the slip is allowed to be. Please add a comment on the capacity for the model to make 'tsunami-earthquake' type scenarios.**

*In our previous studies (i.e. Muhammad et al., 2017 and 2018), we confirmed that the slip distribution parameters (mean and maximum of slip and its spatial distribution) influence the tsunami hazard significantly. We believe that even for the magnitude below M 8.0, the tsunami hazard level from the stochastic*

*earthquake source models may produce a nonnegligible level of tsunami as found in the Mentawai 2010 event.*

*To confirm this, Figure R2-3 shows the coastal height along the coast of western Sumatra (Figure R2-3a) produced from the stochastic earthquake source models for the M 7.75 ruptured in the third segment (i.e. around the Mentawai Islands). The figure shows that the M 7.75 event may produce a significant tsunami height of about 3 m due to the concentrated slip close to the coastal line with a relatively high slip amount (i.e. 1-2 m of mean slip for M 7.75; Muhammad at el., 2018).*

**L317 "... 300 stochastic models are sufficient to simulate stable and consistent tsunami heights and depths" -- I think this must depend on the model region, and what you are interested in. For instance it would not give an accurate representation of the 99.5th percentile. Also for a model where only a very small part of the source-zone could affect the site of interest, one might need to generate many scenarios to get enough relevant scenarios. In summary, I don't think you can refer to stability tests from another study to provide justification for using 300 models in this study. Instead, can you report on a test that is specific to this case?**

*Thank you very much for your valuable comments, and we agree that the number of simulations might differ from region to region. However, we have done a stability check for the stochastic tsunami simulation in the Mentawai-Sunda subduction zones in our previous studies (i.e. Muhammad et al., 2016, 2017, 2018) and confirmed that the 300 model is sufficient to produce a stable tsunami hazard results.*

[Figure]

*Figure R2-5. Convergence of estimated tsunami intensity measures (wave height) as a function of the number of simulations by considering six magnitude scenarios (M = 7.75, M 8.0, M 8.25, M 8.5, M 8.75, and 9.0) using the median (a) and the 95th percentile (b) of the tsunami heights.*

*To confirm that the 300 source model is sufficient in this study, we plot the 95th percentile and the median of tsunami height at a point along the coast of western Sumatra (Point no. 4 – P4 in Figure 2A) against the number of stochastic models from all magnitude scenarios (i.e. M 7.75, M 8.0, M 8.25, M*

*8.5, M 8.75, and M 9.0; see Figure R2-5). The figure confirms that, in general, it takes about 250 stochastic source models to produce a stable tsunami hazard. Hence, it clarifies that the number of stochastic source models used in this study (i.e. 300 models) is sufficient.*

*Therefore, we will add such an explanation in the revised manuscripts once we are invited.*

**L347: "... the final parameter estimates are taken from the maximum a posterior". It would be better to account for the model uncertainty (also in Mw-max, b, etc), which should be substantial given the limited data available to fit the model, and will probably have a substantial impact on the hazard. One way to do this would be to draw a different parameter set for each of the large number of synthetic catalogues that are simulated.**

*We agree with the reviewer's comments and will re-analyze the Bayesian estimation by updating the following steps:*

*(1) The prior setup needs to be updated and adopt the uninformative priors, e.g. as used by Fitzenz et al. (2010).*
*(2) The BPT parameters estimation using the Bayesian approach will be further improved to use both the maximum a posteriori (MAP) and full posterior parametric uncertainty in defining the final BPT parameters to consider the uncertainty in BPT parameter estimation.*

**Section 3.1: As discussed earlier, please comment on why the 'mu-priors' for some segments are so different to the posteriors (little overlap for Fig 9 panels A, I, K). This is surprising given especially considering that the priors were apparently constructed using the same data used for fitting. To me it suggests weaknesses in how the priors were constructed, or some other problem.**

*We thank you for these valuable comments and are aware that the BPT parameter estimation needs to be re-evaluated by considering different prior models. Therefore, we will update the new results once invited to revise the manuscript.*

**L356-360: This is a very clear description of how the catalogue duration was defined. I suggest you move this to the earlier methods section (where I expressed confusion about the method).**

*We will move this section to the methodology as suggested.*

**L360 and Figure 10: Regarding the validation of the annual seismic moment release: Considering that the data was used to fit the model, I don't think the observations/model are particularly consistent on segments 3 and 4. In both cases the observed data exceeds the 90th percentile of the model. Again this seems to suggest some under-estimation in the model, as discussed**

**repeatedly above. Please check that this is all correct following revisions, and if it is, add a comment explaining why this is nonetheless reasonably consistent.**

*Thank you very much for your comments. We have discussed this issue in point number 3 (P3) of responding to your 'High-Level Comments'.*

**L379: Figure 11C is not a strong basis for making a point about which segments rupture more or less, because it is only 1 catalogue. Can you please make a figure that better justifies the points made in this paragraph?**

*We agreed and will update the figure accordingly to clarify our point.*

**Line 447 and Figure 12: The conditional probability of Mw9.0 (if an earthquake occurs) is larger in the time independent case. But I doubt that these results will be robust to parameter uncertainties in the time-dependent model, considering that limited data (10 events, that likely has time-varying completeness) was available to fit its many parameters. This further suggests the importance of considering model parameter uncertainties in the PTHA.**

**Line 451: One factor neglected in this discussion is the effect of using different datasets to fit the 2 models, which could cause differences in the results even if there was no other difference between the two kinds of models. I think the calculations in this paper should be revised so that the time-independent model is informed by the long-term data, and that parameter uncertainties in both the time-independent and time-dependent models are accounted for. In my judgment it is likely that the parameter uncertainties will could lead to differences in the results that are substantially larger than differences between the current time-dependent and time-independent models.**

*Thank you very much for your comments. We have discussed these issues related to the two lines in point number 3 (P3) of responding to your 'High-Level Comments'.*

**References**

Fitzenz, D. D., Ferry, M. A., & Jalobeanu, A. (2010). Long-term slip history discriminates among occurrence models for seismic hazard assessment. Geophysical Research Letters, 37(20), 1–5. https://doi.org/10.1029/2010GL044071.

Goda, K., Yasuda, T., Mori, N., and Maruyama, T. (2016). New scaling relationships of earthquake source parameters for stochastic tsunami simulation. Coastal Eng. J.. doi:10.1142/S0578563416500108.

Griffin, J. D.; Stirling, M. W. & Wang, T. Periodicity and Clustering in the Long-Term Earthquake Record Geophysical Research Letters, American Geophysical Union (AGU), 2020, 47.

Hayes, G.P., Wald, D.J., and Keranen, K. (2009). Advancing techniques to constrain the geometry of the seismic rupture plane on subduction interfaces a priori - higher order functional fits, *Geochem. Geophys. Geosyst.* 10, Q09006, doi:10.1029/2009GC002633.

Hayes, G.P., Wald, D.J., and Johnson, R. L. (2012), Slab1.0: A three-dimensional model of global subduction zone geometries, J. Geophys. Res. 117, B01302, doi:10.1029/2011JB008524.

Horspool, N., Pranantyo, I., Griffin, J., Latief, H., Natawidjaja, D. H., Kongko, W., ... & Thio, H. K. (2014). A probabilistic tsunami hazard assessment for Indonesia. Natural Hazards and Earth System Sciences, 14(11), 3105-3122.

Moernaut, J. (2020, November 1). Time-dependent recurrence of strong earthquake shaking near plate boundaries: A lake sediment perspective. Earth-Science Reviews. Elsevier B.V. https://doi.org/10.1016/j.earscirev.2020.103344.

Natawidjaja, D.H., Sieh, K., Chlieh, M., Galetzka, J., Suwargadi, B.W., Cheng, H., Edwards, R.L., Avouac, J.P., and Ward, S.N. (2006). Source parameters of the great Sumatran megathrust earthquakes of 1797 and 1833 inferred from coral microatolls. J. Geophys. Res. 111, B06403, doi:10.1029/2005JB004025.

Newman, A.V., Hayes, G., Wei, Y., and Convers, J. (2011). The 25 October 2010 Mentawai tsunami earthquake, from real-time discriminants, finite-fault rupture, and tsunami excitation. Geophys. Res. Lett. 38, 1–7. doi:10.1029/2010GL046498.

Muhammad, A., Goda, K., & Alexander, N. (2016). Tsunami hazard analysis of future megathrust sumatra earthquakes in Padang, Indonesia using stochastic tsunami simulation. Frontiers in Built Environment, 2, 33.

Muhammad, A., Goda, K., Alexander, N. A., Kongko, W., & Muhari, A. (2017). Tsunami evacuation plans for future megathrust earthquakes in Padang, Indonesia, considering stochastic earthquake scenarios. Natural Hazards and Earth System Sciences, 17(12), 2245-2270.

Muhammad, A., & Goda, K. (2018). Impact of earthquake source complexity and land elevation data resolution on tsunami hazard assessment and fatality estimation. Computers & geosciences, 112, 83-100.

Philibosian, B., Sieh, K., Avouac, J.P, Natawidjaja, D.H., Chiang, H., Wu, C., Perfettini, H., Shen, C.C., Daryono, M.R., and Suwargadi, B.W. (2014). Rupture and variable coupling behavior of the Mentawai segment of the Sunda megathrust during the super cycle culmination of 1797 to 1833. J. Geophys. Res. Solid Earth. 119, 7258–7287, doi:10.1002/2014JB011200.

Philibosian, B., Sieh, K., Avouac, J. P., Natawidjaja, D. H., Chiang, H. W., Wu, C. C., ... & Wang, X. (2017). Earthquake supercycles on the Mentawai segment of the Sunda megathrust in the seventeenth century and earlier. Journal of Geophysical Research: Solid Earth, 122(1), 642-676.

Satake, K., Nishimura, Y., Putra, P. S., Gusman, A.R., Sunendar, H., Fujii, Y., Sunendar, H., Latief, H., and Yulianto, E. (2013). Tsunami source of the 2010 Mentawai, Indonesia earthquake inferred from tsunami field survey and waveform modeling. Pure Appl. Geophys. 170, 1567–1582. doi:10.1007/s00024-012-0536-y.

Sieh, K., Natawidjaja, D.H., Meltzner, A.J., Shen, C.C., Cheng, H., Li, K.S., Suwargadi, B.W., Galetzka, J., Philibosian, B., and Edwards, R.L. (2008). Earthquake super cycles inferred from sea-level changes recorded in the corals of West Sumatra. Science 322, 1674–1678. doi:10.1126/science.1163589.

Thingbaijam, K. K. S., Mai, P. M., & Goda, K. (2017). New Empirical Earthquake Source-Scaling LawsNew Empirical Earthquake Source-Scaling Laws. Bulletin of the Seismological Society of America, 107(5), 2225-2246.

Weichert, D. H. (1980). Estimation of the earthquake recurrence parameters for unequal observation periods for different magnitudes. Bulletin of the Seismological Society of America, 70(4), 1337-1346.

Williams, R. T., Davis, J. R., & Goodwin, L. B. (2019). Do Large Earthquakes Occur at Regular Intervals Through Time? A Perspective From the Geologic Record. Geophysical Research Letters, 46(14), 8074–8081. https://doi.org/10.1029/2019GL083291.

Yue, H., Lay, T., Rivera, L., Bai, Y., Yamazaki, Y., Cheung, K.F., Hill, E.M., Sieh, K., Kongko, W., and Muhari, A. (2014). Rupture process of the 2010 Mw7.8 Mentawai tsunami earthquake from joint inversion of near-field hr-GPS and teleseismic body wave recordings constrained by tsunami observations. J. Geophys. Res. Solid Earth. 119, 5574–5593. doi:10.1002/2014JB011082.

Zachariasen, J., Sieh, K., Taylor, F.W., Edwards, R.L., and Hantoro, W.S. (1999). Submergence and uplift associated with the giant 1833 Sumatran subduction earthquake: Evidence from coral microatolls. J. Geophys. Res. 104, 895–919.

---

## Author Comment (AC3)

**Response to the comments of Reviewer 3**

The authors would like to thank the reviewers for their constructive and positive reviews of our manuscript. We would like to highlight several essential points that all three reviewers commonly ask:

First, our paper is a manuscript that proposes a new approach integrating the multivariate Bernoulli earthquake occurrence model and stochastic source modelling. As such, a natural scope/application of the new method is the time-dependent Probabilistic Tsunami Hazard Analysis (PTHA). We do not take a strong position as to which time-independent and time-dependent models are appropriate. However, we have noticed that the wordings in our original manuscript in supporting the time-dependent model for PTHA were relatively strong, and hence, once we are invited to revise the manuscript, we will soften our statements and clarify the motivation of the paper.

Second, the justification for the choice of the time-dependent PTHA model was not well presented in the submitted manuscript. We have prepared additional texts both in the introduction and the results and discussion sections (see details comments below) to highlight some more recent studies (i.e. Williams et al., 2019; Griffin et al., 2020; Moernaut 2020). Those studies suggest that the global paleoearthquake records provided empirical support for weakly quasiperiodic earthquake recurrence. Therefore, it can be used to justify the use of renewal models (i.e. Brownian Passage Time (BPT) model) for seismic/tsunami hazard assessment.

Third, this manuscript presents a new alternative to carrying out a time-dependent PTHA as a kind of sensitivity analysis using Indonesia (Sunda) applications. As such, we should recognize the limitations and challenges to be resolved. In our submitted manuscript, we did not provide the limitations of our study highlighted by the three reviewers. These include the 1D approach in earthquake rupture modelling, BPT parameter estimation using the Bayesian approach, and space-time earthquake rupture modelling. Subsequently, we will dedicate a new section explaining our limitations in the revised manuscript.

Fourth, a combination of multiple earthquake occurrence models can be included as a logic-tree approach. However, we did not explore this in this study as it is out of the scope. We stay on our objective: presenting a new method that combines multivariate Bernoulli and stochastic source modelling methods, which we hope to promote many applications in future.

Fifth, the valuable suggestion from the reviewers to re-do the analysis, specifically in the earthquake rupture modelling and Bayesian parameter estimation, is essential. We will re-do the earthquake rupture modelling and Bayesian parameter estimation and include the updated results in the revised manuscript:

For the earthquake rupture modelling:

- The instrumental events (i.e. the 2007a, 2007b and the 2010 events) will be excluded in modelling the time-dependent earthquake rupture.
- The earthquake catalogue data to develop a magnitude-frequency model for the time-independent approach will include the paleogeodetic records provided in Philibosian et al. (2017) by excluding the instrumental events (i.e. the 2007a, 2007b and the 2010 events). Therefore, both the time-dependent and time-independent approaches use similar earthquake data. An updated frequency-magnitude model will then be developed for the time-independent approach by fitting the Gutenberg-Richter (GR) relationship using Weichert (1980) method to treat the varying completeness magnitude. Moreover, the number of low-magnitude scenarios (i.e. < M 7.625) will be reduced by considering only the magnitude of ≥ M 6.0 (in the submitted manuscript, we use the magnitude of ≥ M 5.0).

For BPT parameters estimation using the Bayesian approach:

- The prior setup needs to be updated and adopt the uninformative priors, e.g. as used by Fitzenz et al. (2010).
- The BPT parameters estimation using the Bayesian approach will be further improved to use both the maximum a posteriori (MAP) and full posterior parametric uncertainty in defining the final BPT parameters to consider the uncertainty in BPT parameter estimation.

Furthermore, to respond to the reviewer's comments, we have copied the reviewer's comments and in italicized text.

**Major comments**

**Justification of the choice of a time-dependent approach. A number of recent studies of global paleoearthquake records (Williams et al 2019; Griffin et al 2020; Moernaut 2020) have, to varying degrees, provided empirical support for weakly quasiperiodic earthquake recurrence as a general model, which can be used to justify the use of renewal models for hazard assessment. That said, the Mentawai record of Philibosian et al (2017) looks to be more random than quasiperiodic in the analysis presented by Griffin et al (2020), although perhaps a different result might be obtained using the segmentation model presented here. The posterior BPT parameter estimates given for each segment are also relevant – some give values of alpha ~1 (segments 2, 3 and 4), implying random recurrence (i.e. Poisson), while others are ~0.6 (segments 1, 5 and 6), implying moderately quasiperiodic recurrence. So, I think some comment needs to be made here that:**

1. **At a global scale there is empirical support for weakly quasiperiodic earthquake recurrence as a general model (see Griffin et al 2020);**

2. **Excluding the hypothesis at the individual fault level is difficult, particularly for short records (Williams et al 2019; Griffin et al 2020)**
3. **The data from Philibosian et al (2017) is somewhat equivocal about whether earthquake recurrence here is truly time-dependent, and the Poisson hypothesis cannot be confidently excluded using these data. But the global studies mentioned above suggest it is not unreasonable to assume time-dependence as a hypothesis.**

**The discussion section of the paper could then discuss the implications of this assumption in light of the different values of alpha obtained for each segment.**

*Thank you very much for these valuable comments. We will revise the manuscript by adding the following text in two different sections to justify the chosen model (i.e. BPT model) once we are invited to revise it.*

**In the Introduction section:**

*In general, the earthquake rupture can be modelled using two approaches: Poisson and non-Poisson. The Poisson approach employs a memory-less Poisson process for long-term hazard assessment and is commonly adopted for earthquake rupture modelling (e.g. Burroughs and Tebbens, 2005; Tinti et al., 2005; Orfanogiannaki and Papadopoulos, 2007). However, assuming a lack of memory between major earthquake occurrences is often viewed as a first approximation, inconsistent with the physics of elastic rebound (Reid, 1911; Anagnos and Kiremidjian, 1984; Berryman et al., 2012). As a result, many studies adopted a renewal model of earthquake occurrence, i.e. non-Poisson model (Matthews et al., 2002; Zhuang et al., 2012; Field et al., 2014; Williams et al., 2019; Griffin et al., 2020) to carry out a time-dependent earthquake rupture modelling. More recent studies using global paleoearthquake records (i.e. Williams et al., 2019; Griffin et al., 2020; Moernaut 2020) showed that large earthquakes in the subduction zones recur more regularly than expected from exponentially distributed interevent times (i.e., a Poisson process). Specifically, the earthquake recurrence in the Mentawai Sunda subduction zone is categorized as more like a supercycle type (i.e. a combination of large gaps and clusters), demonstrating that successive large earthquakes are dependent on each other (Salditch et al. 2020). Therefore, this study adopts the renewal model (i.e. BPT distribution) for earthquake rupture modelling in the Mentawai segment of the Sunda subduction zone.*

**In the results section of 'Bayesian parameter estimation':**

***Error! Reference source not found.*** *illustrates the MCMC results with the priors and posteriors of $\mu$, $\alpha$, and $\gamma$ for all segments, whilst the final parameter estimates are taken from the maximum a posteriori (MAP; Table 3). The figure shows that, in general, the available earthquake data can effectively reduce the parametric uncertainty of the priors, in particular for the $\mu$ parameter in*

*segments 2 to 4. The median interarrival times of the central segments (i.e. segments 2 to 4) are about 40 years, while the interarrival times in the remaining segments are greater by more than 50%. The uncertainties of the parameters for the interarrival times of segments 1, 5, and 6 are large because few ruptures have occurred in those segments (see **Error! Reference source not found.**). Moreover, the data dispersion of the central segments is greater than in others, resulting in a higher coefficient of variation. Moreover, the values of $\alpha$ for each segments are varied depending on the segments. Segments 2, 3, and 4 have an $\alpha$ of ~1 implying random recurrence (i.e. Poisson). On the other hand, the values of $\alpha$ for segments 1, 5, and 6 are ~0.6 showing a moderately quasiperiodic recurrence. Such results are generally consistent with the findings from the recent studies using global paleoearthquake records, including the data for the Mentawai segment of the Sunda subduction zone (Williams et al. 2019; Griffin et al. 2020; Moernaut 2020). The study suggests that the earthquake recurrence for the Mentawai-Sunda zone can be time-dependent, but the Poisson hypothesis cannot be excluded to model the future earthquake rupture.*

**In estimating parameters for the BPT distribution, the authors use the data to estimate the prior distribution of mu, before then using the same data to calculate the posterior probability distribution of mu. This is incorrect. I would suggest using an uninformative prior (e.g. as used by Fitzenz et al 2010). An alternative approach could be to use an informative prior for mu based on the slip rate (e.g. as determined from geodesy), but this may become complex (e.g. due to having to estimate coupling of the fault). The 450 year long record is short for accurately estimate model parameters. This is, of course, what a Bayesian approach should be helping with, but needs more care about the choice of priors.**

*We agree with the reviewer's comments and will re-analyze the Bayesian estimation by updating the following steps:*

*(1) The prior setup needs to be updated and adopt the uninformative priors, e.g. as used by Fitzenz et al. (2010).*
*(2) The BPT parameters estimation using the Bayesian approach will be further improved to use both the maximum a posteriori (MAP) and full posterior parametric uncertainty in defining the final BPT parameters to consider the uncertainty in BPT parameter estimation.*

**I am also concerned that fitting the model parameters to each segment individually is problematic. Later you consider multi-segment ruptures, and it is not clear how all this fits together. Do the recurrence statistics obtained from the sum of all synthetic ruptures across all segments match the recurrence statistics from the sum of all historic/paleo ruptures in your data? Checking this could be a good test for your model.**

*Thank you very much for raising this issue.*

*In our study, the rupture is modelled based on the integration of temporal and spatial interaction shown by $\boldsymbol{p_t}$ and $\Sigma$ in Equation (5) of the manuscript. $\boldsymbol{p_t}$ is a vector of the marginal probability of rupture on the $k$-th segment in year $t$ given the time since the last rupture ($T_t$). $\Sigma$ is a 6-by-6 covariance matrix describing the spatial correlation of ruptures on the segments. Subsequently, the rupture is modelled for each segment each year, where the spatial correlation, $\Sigma$, constrains the extension of the rupture. Such a spatial correlation allows us to model the rupture of the whole segment.*

*Moreover, we will check the consistency of our rupture modelling results regarding the recurrence statistics once we are invited to revise the manuscript. Currently, we have not explored such a result consistency because we will re-do the rupture modelling as described in the introduction of our response.*

**Also related to parameter estimation, some of the posterior histograms seem a bit spiky; does this improve if the number of samples is increased beyond 10,000?**

*The posterior distribution does not change much even when we increase the number of samples. The spiky at the posterior distribution is generally found at the segment (e.g., segments 1 and 6) where the past earthquake data is insufficient (only one or two events rupture in those segments. Once we update the Bayesian parameter estimation in the revised manuscript (if we are invited to revise it), we will update the evaluation of the posterior distribution and the BPT parameter estimation.*

**Spatio-temporal completeness of the paleo record compared with the instrumental record is an issue that I think could lead to biases in the parameter estimates. It is very unlikely that events similar to the Mw 7.8 2010 Mentawai event would be visible in the coral record; this event occurred near the trench and caused <4 cm subsidence on the Mentawai Islands as measured with GPS (Hill et al 2012). Related to the above, the Mmin of 7.6 (L129), while reasonable from a tsunami hazard assessment perspective, would mean that you are modelling events that are unlikely to be present in the paleoearthquake record. I am unsure of how the frequency of these events could be determined in the time-dependent approach. Therefore it seems likely in your current approach that smaller events are missed in the paleoearthquake record, therefore affecting the recurrence model parameters.**

*Thank you very much for these valuable suggestions.*

*We realized that we have a few technical weaknesses in developing the earthquake rupture modelling, including:*

*(1) Use different catalogue data to model the magnitude frequency relationship between the time-dependent and time-independent models. The time-independent model adopts only the earthquake catalogue data from 1970, whilst the time-dependent data consider solely the paleogeodetic records from the sixteenth century.*

*(2) Consider a large number of small earthquakes (i.e. M 5.5 – M 6.0) for the time-independent model that is well below the minimum magnitude considered in this study (i.e. M 7.625).*

*(3) Include the instrumental events (i.e. the 2007a, 2007b and the 2010 events) for the time-dependent model that the paleogeodetic records can not detect.*

*Subsequently, the following revisions will be conducted in the revised manuscript once we are invited:*

*First, the instrumental events (i.e. the 2007a, 2007b and the 2010 events) will be excluded in modelling the time-dependent earthquake rupture.*

*Second, the earthquake catalogue data to develop a magnitude-frequency model for the time-independent approach will include the paleogeodetic records provided in Philibosian et al. (2017) by excluding the instrumental events (i.e. the 2007a, 2007b and the 2010 events). Therefore, both the time-dependent and time-independent approaches use similar earthquake data. An update frequency-magnitude model will then be developed for the time-independent approach by fitting the Gutenberg-Richter (GR) relationship using Weichert (1980) method to treat the varying completeness magnitude. Moreover, the number of low-magnitude scenarios (i.e. < M 7.625) will be reduced by considering only the magnitude of ≥ M 6.0 (in the submitted manuscript, we use the magnitude of ≥ M 5.0).*

*Consequently, we perform the initial modelling of the magnitude frequency relationship by considering the new earthquake catalogue (i.e. integration catalogue) to understand the difference in the magnitude frequency distribution before and after the integration of paleogeodetic records. The results show that the b value produced from the Weichert (1980) approach is far less than the previous model, i.e. $b = 0.72$ for the Weichert model vs $b = 1.05$ for the catalogue excluding the paleogeodetic record. Furthermore, we plot the probability distribution of magnitude from the non-integrated (excluding the paleogeodetic records) and the integrated catalogues (including the paleogeodetic records) as shown in Figure R3-1. The figure clearly shows a significant change in magnitude probability for low (< M 8.0) and high (> M 8.3) scenarios. The probability of low magnitude in the integrated catalogue is almost twice smaller than the non-integrated catalogue. In contrast, the probability of the high magnitude of the integrated catalogue is twice higher than that of the-non integrated catalogue. Such a change will definitely influence the final hazard curve of the time-independent model.*

[Figure]

*Figure R3-1. Probability magnitude distribution produced from the non-integrated catalogue (a) and the integrated catalogue (b).*

*The modified results for comparing time-dependent and time-independent PTHA will be included in the revised manuscript, specifically in the methodology, results and discussions sections. We will run the new earthquake rupture modelling for the time-dependent case by excluding the instrumental events and update the results accordingly once we are invited to revise the manuscript.*

**The 1D rupture segmentation is a problem for tsunami hazard assessment, as the resulting tsunami size depends so significantly on the depth of rupture. Compare the 2007 Bengkulu earthquakes (Mw 8.4 and 7.9), that were down-dip of the trench and did not generate a significant tsunami, with the 2010 Mentawai earthquakes (Mw7.8), which occurred near the trench and did generate a significant tsunami. It is not clear whether such events are discriminated by the stochastic modelling approach with 1D segmentation – it seems they probably aren't, but I may not be understanding correctly.**

*Thank you very much for your valuable comment.*

*The 2D segmentation can also be possibly represented by our current setup. The first direction is along strike segmentation whereas the asperity region can be regarded as a along width segmentation as large concentration of slips is allowed in this area. This was initially considered to accommodate notable observations from the recent tsunami events, including 2011 Tohoku – large concentration of slips along the trench line. In reality, this may capture various geological environments – such as outer wedge rupture. However, such an explanation has not been included in the submitted manuscript and we will include this in the revised manuscript once we are invited to revise it.*

*On the other hand, In our setup, the strike and dip angles of the fault-plane of tsunamigenic source models are typically 296º to 326º and 7º to 19º, respectively. These values are comparable to the slab models for the Sunda*

*subduction zone produced by the USGS (Hayes et al., 2009, 2012). The top edge of the fault plane is located at a depth of 3 km. This depth is consistent with the past Mentawai finite-fault models developed for the 2010 Mentawai tsunamigenic earthquakes and the twin events of the 1797 and 1833, which have the top edge depth between 2 km and 5 km (Newman et al., 2011; Satake et al., 2013; Philibosian et al., 2014; Yue et al., 2014).*

*We have not explained this clearly in the manuscript, and we will include it in the manuscript once we are invited to revise the manuscript.*

**A related problem is low-rigidity near the trench and its tsunamigenic potential, as in the 2010 Mentawai tsunami? How might the assumption of constant (and relatively high) rigidity (L309-310) bias your tsunami hazard results?**

*In our approach, we allow large concentration of slips in the asperity region. This was initially considered to accommodate notable observations from the recent tsunami events, including 2011 Tohoku – large concentration of slips along the trench line. In reality, this could capture various geological environments – such as outer wedge rupture. The large slip 'could' reflect low rigidity of the outer wedge of the accretionary prism or others.*

*Moreover, It is also important to note that all inversion studies in this modelling framework are typically done by considering a (hard) constant ridigity, to match the predictions with the observations. Consequently, the slips along the trench are large. Therefore, the stochastic earthquake source modelling used in this study with a constant rigidity and allowable large slip concentration in the asperity region may still capture realistic tsunami hazard levels in the region of interest.*

**The maximum magnitude of 9.0 seems too low, which seems related to the segmentation model. If the potential for ruptures connecting with other segments of the Sunda Subduction Zone is considered, then larger Mmax values are justified. Significantly larger Mmax's were used in Horspool et al (2014). Even if the paleoearthquake record for the past 450 years suggests events haven't exceeded Mw 9.0, we also don't expect these magnitude events to occur all that often. So allow for the possibility that they are missing from the record.**

*Thank you very much for your comments.*

*In this study, to integrate the spatio-temporal time-dependent earthquake rupture modelling, we adopt a 1D along-strike distance discretized into six segments. Each segment represents the smallest area that may rupture in tsunamigenic earthquakes. Such a 1D along-strike model is developed based*

*on the past fault rupture, and the total distance of these 6 segments (i.e. 600 km) is used to represent the maximum magnitude scenario.*

*The maximum magnitude scenario was selected based on geodetic, paleo-geodetic, and paleo-tsunami studies (Zachariasen et al., 1999; Natawidjaja et al., 2006; Sieh et al., 2008). Those studies indicated that the accumulated slip in the Mentawai segment of the Sunda subduction zone might generate tsunamigenic earthquakes ranging from M 8.8 to M 9.1. Specifically, coral microatoll samples from 21 sites along 600 km of the Mentawai region were used to constrain the dates, spatial extents and approximate earthquake source models for tsunami generation (Natawidjaja et al., 2006; Shieh et al., 2008; Philibosian et al., 2014, 2017). In our study, the tsunami simulation can not be extended into more than six segments (600 km) due to the availability of the coral microatoll samples beyond these six segments. However, this 600-km segment with a total area of 150,000 km², i.e. 600 km (length) × 250 km (width) is still consistent with the scaling relationships (e.g. Goda et al. (2016) and Thingbaijam et al. (2017)). Figure R3-2 presents the relationship between the fault areas and magnitude based on the Goda et al. (2016) and Thingbaijam et al. (2017) scaling relationships. The figure shows that the maximum fault areas used in this study represent the maximum magnitude of between M 9.0-9.2. Moreover, we allow the model to have a maximum magnitude up to about M 9.1 (+/- M 0.1 of M 9.0) in developing the stochastic earthquake source model, and hence, such a number is still sufficient to represent the maximum magnitude event that may occur in the Mentawai segment of the Sunda subduction zone. Therefore, it can still be adopted to constrain the boundaries of finite fault models developed in this work.*

[Figure]

*Figure R3-2. Fault areas and M relationships based on the Goda et al. (2016) and Thingbaijam et al. (2017) relationships.*

*More importantly, we are aware that this is one of the limitations of our current work. Therefore, we will emphasize this issue in the following revised manuscript once we are invited to revise it.*

**Some area of the coast of Padang show zero probability of inundation (Figure 17), while in others the potential inundation extent extends quite a way inland. This raises some significant concerns for me about the quality of the inundation modelling and/or the elevation data used, given how low-lying the coast is in this area. If only SRTM data was used, this could significantly underestimate inundation extent (see Griffin et al 2015, Figure 8). Are buildings included in the elevation model?**

*Thank you very much for your comments.*

*The building is not included in our model, and we don't solely use the SRTM data. For Padang areas, we use the local DEM5 and Bathy5 for the DEM and bathymetry data (Taubenbock et al., 2009; Schlurmann et al., 2010). Outside the Padang region, we adopt the SRTM dataset.*

[Figure]

*Figure R3-3. Tsunami hazard map (right panel) in Padang produced from one of the M 9.0 stochastic source models (i.e. model 9; left panel).*

*The inundation results in Figure 17 are developed based on the time-dependent and time-independent methods for two different probabilities (10% and 2%) in 50 years. Using such an approach, we take the tsunami hazard at an interest probability level of 0.02 and 0.1. Therefore, the extension of inundation inland is zero in some of the Padang's areas compared to our previous tsunami hazard maps in Padang (i.e. in Muhammad et al., 2017 and 2018). Suppose we consider only the tsunami hazard maps from one example of generated stochastic source models. We may see the extension of the tsunami hazard maps to be further inland, as shown in Figure R3-3.*

<h1 align="center">Detailed comments</h1>

**L15: Suggest change 'A total of >' to 'More than'**

*We will update the manuscript based on your suggestions.*

**L18: Forecast periods begin in what year?**

*The forecast began in 2021. We will add such an explanation in the revised manuscript.*

**L136: Choice of BPT is fine, but hasn't really been justified here. Why is this chosen over lognormal, Weibull or Gamma? Some of your justification seems to be presented later in Section 2.2.**

*In the revised manuscript, we will add the following texts (in the introduction) once we are invited to revise the manuscript:*

> *In general, the earthquake rupture can be modelled using two approaches: Poisson and non-Poisson. The Poisson approach employs a memory-less Poisson process for long-term hazard assessment and is commonly adopted for earthquake rupture modelling (e.g. Burroughs and Tebbens, 2005; Tinti et al., 2005; Orfanogiannaki and Papadopoulos, 2007). However, assuming a lack of memory between major earthquake occurrences is often viewed as a first approximation, inconsistent with the physics of elastic rebound (Reid, 1911; Anagnos and Kiremidjian, 1984; Berryman et al., 2012). As a result, many studies adopted a renewal model of earthquake occurrence, i.e. non-Poisson model (Matthews et al., 2002; Zhuang et al., 2012; Field et al., 2014; Williams et al., 2019; Griffin et al., 2020) to carry out a time-dependent earthquake rupture modelling. More recent studies using global paleoearthquake records  (i.e. Williams et al., 2019; Griffin et al., 2020; Moernaut 2020) showed that large earthquakes in the subduction zones recur more regularly than expected from exponentially distributed interevent times (i.e., a Poisson process). Specifically, the earthquake recurrence in the Mentawai Sunda subduction zone is categorized as more like a supercycle type (i.e. a combination of large gaps and clusters), demonstrating that successive large earthquakes are dependent on each other (Salditch et al. 2020). Therefore, this study adopts the renewal model (i.e. BPT distribution) for earthquake rupture modelling in the Mentawai segment of the Sunda subduction zone.*

**L174: Several thousand years**

*We thank you for this comment and will revise the manuscript based on your suggestions.*

**L185: Perhaps rephrase as 'reflects the expectations of elastic rebound theory', or similar.**

*We will revise the manuscripts based on your suggestions.*

**L192: Should probably cite others who've used Bayesian approaches to fitting time-dependent models to earthquake records, in particular Rhoades et al (1994) and Fitzenz et al (2010).**

*We will cite the mentioned literature in the revised manuscript.*

**L197 and Table 1: These should not be referred to as tsunamigenic. For half of them we have no information on whether a tsunami was generated; coseismic deformation on the Mentawai Islands observed in coral paleogeodetic records suggests they probably were, but we don't actually know.**

*In the revised manuscript, we will update the term of tsunamigenic to earthquake event.*

**L324. Please give a link or citation for DEM5 and Bathy5.**

*We will add the following citations in the revised manuscript: Taubenbock et al., 2009 and Schlurmann et al., 2010.*

**L332: Might be a typo here – Griffin et al (2016) used a Manning's roughness of 0.036 as a conservative minimum for land (grassland; for the Mentawai Islands). For the urban context here, 0.06 may be reasonable, e.g. Griffin et al (2015) suggested a Manning's roughness of 0.08 for the city of Padang. See also Kaiser et al (2011) for a discussion of choice of Mannings n.**

*Thank you very much for noticing such a typo. We will update the typo and include some essential references here in the revised manuscript.*

**Ling 501-502: The time-independent model has too low an Mmax (9.0) to be considered worst-case. See earlier comments about choice of Mmax.**

*Thank you very much for the comment. We have answered this in the major comment point 8.*

**Table 1: Change Shieh to Sieh.**

*We will revise the manuscripts based on your suggestions.*

**Figure 3, and also in the text. I do not think the term 'occurred' scenarios is the best terminology. These are modelled scenarios that have not actually occurred.**

*Thank you very much for this comment. The term 'occurred' scenarios will further be updated to the 'simulated scenarios' in the revised manuscript.*

**References**

Fitzenz, D. D., Ferry, M. A., & Jalobeanu, A. (2010). Long-term slip history discriminates among occurrence models for seismic hazard assessment. Geophysical Research Letters, 37(20), 1–5. https://doi.org/10.1029/2010GL044071.

Goda, K., Yasuda, T., Mori, N., and Maruyama, T. (2016). New scaling relationships of earthquake source parameters for stochastic tsunami simulation. Coastal Eng. J.. doi:10.1142/S0578563416500108.

Griffin, J. D.; Stirling, M. W. & Wang, T. Periodicity and Clustering in the Long-Term Earthquake Record Geophysical Research Letters, American Geophysical Union (AGU), 2020, 47.

Hayes, G.P., Wald, D.J., and Keranen, K. (2009). Advancing techniques to constrain the geometry of the seismic rupture plane on subduction interfaces a priori - higher order functional fits, *Geochem. Geophys. Geosyst.* 10, Q09006, doi:10.1029/2009GC002633.

Hayes, G.P., Wald, D.J., and Johnson, R. L. (2012), Slab1.0: A three-dimensional model of global subduction zone geometries, J. Geophys. Res. 117, B01302, doi:10.1029/2011JB008524.

Horspool, N., Pranantyo, I., Griffin, J., Latief, H., Natawidjaja, D. H., Kongko, W., ... & Thio, H. K. (2014). A probabilistic tsunami hazard assessment for Indonesia. Natural Hazards and Earth System Sciences, 14(11), 3105-3122.

Moernaut, J. (2020, November 1). Time-dependent recurrence of strong earthquake shaking near plate boundaries: A lake sediment perspective. Earth-Science Reviews. Elsevier B.V. https://doi.org/10.1016/j.earscirev.2020.103344.

Natawidjaja, D.H., Sieh, K., Chlieh, M., Galetzka, J., Suwargadi, B.W., Cheng, H., Edwards, R.L., Avouac, J.P., and Ward, S.N. (2006). Source parameters of the great Sumatran megathrust earthquakes of 1797 and 1833 inferred from coral microatolls. J. Geophys. Res. 111, B06403, doi:10.1029/2005JB004025.

Newman, A.V., Hayes, G., Wei, Y., and Convers, J. (2011). The 25 October 2010 Mentawai tsunami earthquake, from real-time discriminants, finite-fault rupture, and tsunami excitation. Geophys. Res. Lett. 38, 1–7. doi:10.1029/2010GL046498.

Muhammad, A., Goda, K., & Alexander, N. (2016). Tsunami hazard analysis of future megathrust sumatra earthquakes in Padang, Indonesia using stochastic tsunami simulation. Frontiers in Built Environment, 2, 33.

Muhammad, A., Goda, K., Alexander, N. A., Kongko, W., & Muhari, A. (2017). Tsunami evacuation plans for future megathrust earthquakes in Padang, Indonesia, considering stochastic earthquake scenarios. Natural Hazards and Earth System Sciences, 17(12), 2245-2270.

Muhammad, A., & Goda, K. (2018). Impact of earthquake source complexity and land elevation data resolution on tsunami hazard assessment and fatality estimation. Computers & geosciences, 112, 83-100.

Philibosian, B., Sieh, K., Avouac, J.P, Natawidjaja, D.H., Chiang, H., Wu, C., Perfettini, H., Shen, C.C., Daryono, M.R., and Suwargadi, B.W. (2014). Rupture and variable coupling behavior of the Mentawai segment of the Sunda megathrust during the super cycle culmination of 1797 to 1833. J. Geophys. Res. Solid Earth. 119, 7258–7287, doi:10.1002/2014JB011200.

Philibosian, B., Sieh, K., Avouac, J. P., Natawidjaja, D. H., Chiang, H. W., Wu, C. C., ... & Wang, X. (2017). Earthquake supercycles on the Mentawai segment of the Sunda megathrust in the seventeenth century and earlier. Journal of Geophysical Research: Solid Earth, 122(1), 642-676.

Satake, K., Nishimura, Y., Putra, P. S., Gusman, A.R., Sunendar, H., Fujii, Y., Sunendar, H., Latief, H., and Yulianto, E. (2013). Tsunami source of the 2010 Mentawai, Indonesia earthquake inferred from tsunami field survey and waveform modeling. Pure Appl. Geophys. 170, 1567–1582. doi:10.1007/s00024-012-0536-y.

Schlurmann, T., Kongko, W., Goseberg, N., Natawidjaja, D. H., & Sieh, K. (2010). Near-field tsunami hazard map Padang, West Sumatra: Utilizing high resolution geospatial data and reseasonable source scenarios. In Proceedings of the Coastal Engineering Conference (2010). Reston: American Society of Civil Engineers.

Sieh, K., Natawidjaja, D.H., Meltzner, A.J., Shen, C.C., Cheng, H., Li, K.S., Suwargadi, B.W., Galetzka, J., Philibosian, B., and Edwards, R.L. (2008). Earthquake super cycles inferred from sea-level changes recorded in the corals of West Sumatra. Science 322, 1674–1678. doi:10.1126/science.1163589.

Taubenböck, H., Goseberg, N., Setiadi, N., Lämmel, G., Moder, F., Oczipka, M., ... & Klein, R. (2009). " Last-Mile" preparation for a potential disaster–Interdisciplinary approach towards tsunami early warning and an evacuation information system for the coastal city of Padang, Indonesia. Natural Hazards and Earth System Sciences, 9(4), 1509-1528.

Thingbaijam, K. K. S., Mai, P. M., & Goda, K. (2017). New Empirical Earthquake Source-Scaling LawsNew Empirical Earthquake Source-Scaling Laws. Bulletin of the Seismological Society of America, 107(5), 2225-2246.

Weichert, D. H. (1980). Estimation of the earthquake recurrence parameters for unequal observation periods for different magnitudes. Bulletin of the Seismological Society of America, 70(4), 1337-1346.

Williams, R. T., Davis, J. R., & Goodwin, L. B. (2019). Do Large Earthquakes Occur at Regular Intervals Through Time? A Perspective From the Geologic Record.

Geophysical Research Letters, 46(14), 8074–8081. https://doi.org/10.1029/2019GL083291.

Yue, H., Lay, T., Rivera, L., Bai, Y., Yamazaki, Y., Cheung, K.F., Hill, E.M., Sieh, K., Kongko, W., and Muhari, A. (2014). Rupture process of the 2010 Mw7.8 Mentawai tsunami earthquake from joint inversion of near-field hr-GPS and teleseismic body wave recordings constrained by tsunami observations. J. Geophys. Res. Solid Earth. 119, 5574–5593. doi:10.1002/2014JB011082.

Zachariasen, J., Sieh, K., Taylor, F.W., Edwards, R.L., and Hantoro, W.S. (1999). Submergence and uplift associated with the giant 1833 Sumatran subduction earthquake: Evidence from coral microatolls. J. Geophys. Res. 104, 895–919.